# ClimateSet: A Large-Scale Climate Model Dataset for Machine Learning

**Julia Kaltenborn**[*]
Mila Quebec AI Institute
& McGill University

**Charlotte Emilie Elektra Lange**
Mila Quebec AI Institute
& University of Osnabrück

**Venkatesh Ramesh**
Mila Quebec AI Institute
& University of Montreal

**Philippe Brouillard**
Mila Quebec AI Institute
& University of Montreal

**Yaniv Gurwicz**
Intel Labs

**Chandni Nagda**
University of Illinois at
Urbana-Champaign

**Jakob Runge**
German Aerospace Center
& Technische Universität Berlin

**Peer Nowack**
Karlsruhe Institute
of Technology

**David Rolnick**
Mila Quebec AI Institute
& McGill University

## Abstract

Climate models have been key for assessing the impact of climate change and simulating future climate scenarios. The machine learning (ML) community has taken an increased interest in supporting climate scientists' efforts on various tasks such as climate model emulation, downscaling, and prediction tasks. Many of those tasks have been addressed on datasets created with single climate models. However, both the climate science and ML communities have suggested that to address those tasks at scale, we need large, consistent, and ML-ready climate model datasets. Here, we introduce ClimateSet, a dataset containing the inputs and outputs of 36 climate models from the Input4MIPs and CMIP6 archives. In addition, we provide a modular dataset pipeline for retrieving and preprocessing additional climate models and scenarios. We showcase the potential of our dataset by using it as a benchmark for ML-based climate model emulation. We gain new insights about the performance and generalization capabilities of the different ML models by analyzing their performance across different climate models. Furthermore, the dataset can be used to train an ML emulator on several climate models instead of just one. Such a "super-emulator" can quickly project new climate change scenarios, complementing existing scenarios already provided to policymakers. We believe ClimateSet will create the basis needed for the ML community to tackle climate-related tasks at scale.

## 1 Introduction

Climate change poses a significant and increasing threat to humans and the environment. Understanding and projecting future climate scenarios is essential to mitigating and adapting to climate change. Those future climate scenarios - the "Shared Socioeconomic Pathways" (SSP) [2] are determined by climate forcer emissions and depend on socioeconomic decisions made by humanity. Here, the term "climate forcers" refers to greenhouse gases (GHG), aerosols, and aerosol precursors, among others.

---

[*]Address correspondence to `julia.kaltenborn[at]mail.mcgill.ca`

[2]For readers who are new to climate modeling terminology, we recommend the glossary by IPCC (2022), also available on `https://www.ipcc.ch/sr15/chapter/glossary/`.

37th Conference on Neural Information Processing Systems (NeurIPS 2023) Track on Datasets and Benchmarks.

To navigate citizens' future in a changing climate, policymakers rely heavily on future simulations of our climate, summarized in reports for the Intergovernmental Panel on Climate Change (IPCC), e.g. Arias et al. (2021)). Those simulations are traditionally created by climate models, which are collected in the Coupled Model Intercomparison Project (CMIP; recently in Phase 6). Climate models are based on physical parameters, equations, and coupling mechanisms that describe the climate and Earth system; these models simulate the climate under different forcing scenarios, e.g. varying greenhouse gas emissions. However, even on top high-performance computing (HPC) clusters, typical climate model simulations take months to run (Balaji et al., 2017).

The machine learning (ML) community has taken increased interest in supporting the climate science community in their efforts to scale and accelerate climate-related modeling tasks. These tasks include climate emulation/projection, downscaling, and general prediction tasks. Climate-related tasks also pose interesting ML challenges due to the high dimensionality of the data, relatively low sample size, and inherent distribution shifts within the data. When approaching those climate-related tasks, ML models have typically leveraged one climate model (Watson-Parris et al., 2022; Cachay et al., 2021; Mansfield et al., 2020; Krasnopolsky et al., 2013; Castruccio et al., 2014; Holden and Edwards, 2010; Beusch et al., 2020), and only rarely several climate models (Nguyen et al., 2023; Yu et al., 2022). This runs counter to the standard practice in ML of leveraging massive datasets. This discrepancy may be due to the difficulty in retrieving, preprocessing, and handling climate data correctly without significant domain knowledge. Indeed the ML community has experienced difficulties in retrieving data of several climate models Nguyen et al. (2023) and making them consistent Busecke and Abernathey (2020). Those data challenges might limit the community's ability to contribute to climate-related modeling tasks.

While the need for a consistent, easy-to-retrieve, and large climate model dataset to train ML models is currently unmet, it has been addressed in parts. For example, these desired datasets can be found for weather (WeatherBench by Rasp et al. (2020)) and satellite data (EarthNet2021 by Requena-Mesa et al. (2021)) rather than climate data. The access to large-scale weather data enabled the development of large ML weather forecasting models (Lam et al., 2022; Bi et al., 2022; Pathak et al., 2022; Gao et al., 2022). Additionally, ClimaX (Nguyen et al., 2023) – the first climate and weather-related large-scale model – relies primarily on weather data. However, we cannot solely use weather data capturing "the past", to address climate change related questions concerning "the future". Extrapolation into such a future is difficult for ML models, especially under strong distribution shifts in space and time. Thus, large-scale and consistent ML datasets are needed not only for weather, but also for climate. Efforts have been made to provide such data: xmip (Busecke and Abernathey, 2020) provides the tools to create more consistent climate model data, however, it does not address all inconsistencies and e.g. cannot align the different temporal and spatial resolutions among climate data. ClimateBench (Watson-Parris et al., 2022) provides a consistent, ML-ready dataset for climate emulation. A drawback to this dataset is that it provides only one climate model. Therefore, it does not capture the multi-model uncertainty that is essential for informing policy making, and is limited in the amount of training data it can provide to ML tasks. Refer to Appendix B for a more comprehensive overview of related ML-datasets. The need expressed by both the ML and climate science communities (Dueben et al., 2022; Runge et al., 2019; Mansfield et al., 2020; Watson-Parris, 2021; Chantry et al., 2021) for a consistent, large, and ML-ready dataset has not yet been addressed jointly for climate data.

Here, we introduce ClimateSet – a consistent, multi-climate-model dataset. We showcase the value of the dataset for the task of climate emulation; however, the dataset can also be used for a wide variety of other tasks. Our main contributions are:

- We introduce the **ClimateSet data pipeline**, which can be used to retrieve and preprocess climate model data from CMIP6 (climate model outputs) and Input4MIPs (climate model inputs) for climate-related ML tasks.

- We use this pipeline to build a **core ClimateSet dataset** with outputs of 36 climate models; and inputs for the emission fields of 4 different Shared Socioeconomic Pathway (SSP) scenarios and historical data.

- We use ClimateSet to compare state-of-the-art ML methods across different climate models on a **climate model emulation** task. We emulate temperature and precipitation responses to climate forcers, obtaining results that are both qualitatively different and more reliable than was possible in previous work.

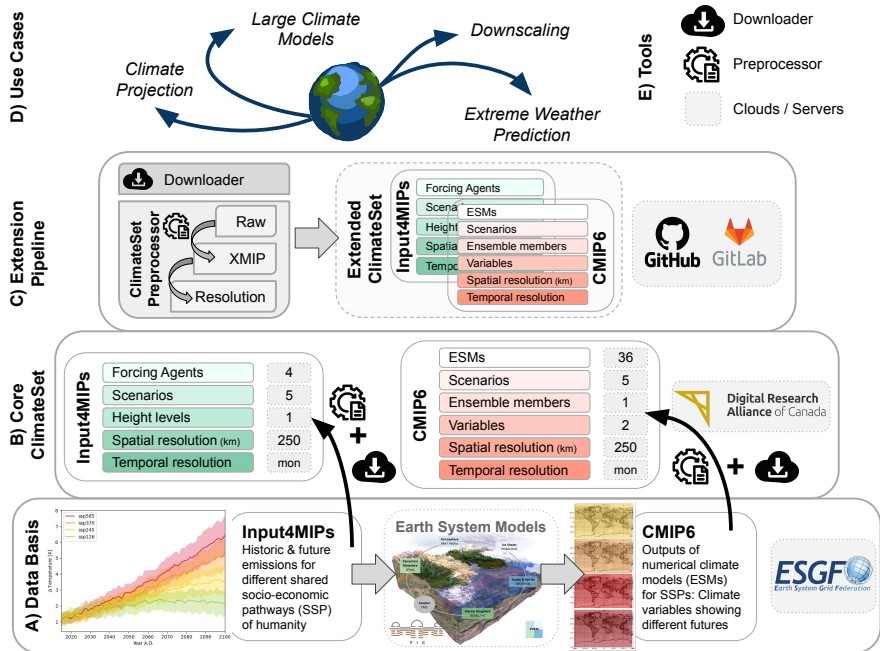

Figure 1: ClimateSet. A) ClimateSet builds on the Input4MIPs and CMIP6 datasets made available through multiple climate modeling teams on the Earth System Grid Federation (ESGF) servers. B) ClimateSet consists of a preprocessed, ML-ready core dataset that includes inputs and outputs for 5 scenarios, 4 climate forcing agents, 2 climatic variables (temperature and precipitation), for a set of 36 climate models. It is currently made publicly available through the Digital Research Alliance of Canada. C) The core dataset can be extended to include different variables, height levels, ensemble members, scenarios and any other information made available from climate models on the CMIP6 server of ESGF. The downloader and preprocessing pipelines are available on our GitHub repository. D) Potential use cases for ClimateSet range from climate projection, climate data downscaling, extreme weather prediction in different warming scenarios, to large ML climate models. E) The main tools provided through ClimateSet are the downloader and the preprocessor to make the climate model data consistent with each other. For further information visit `https://climateset.github.io`. (The 3D Earth System Model visualization was created by Boris Sakschewski, used with permission).

## 2 ClimateSet

The core dataset of ClimateSet consists of 36 climate models and their corresponding greenhouse gases, aerosols and aerosol precursor emission inputs for five different scenarios. The core dataset can be extended with the pipeline we provide. For an overview of ClimateSet refer to Fig. 1. The dataset and the pipeline are both publicly available on `https://climateset.github.io`. ClimateSet serves two main purposes: (1) Providing the amount of training data needed for large-scale ML models; and (2) capturing the projection uncertainty across climate models that is key for climate policy making. Both purposes can only be fulfilled by a dataset containing several climate models. The following describes the core dataset, how the data was collected, its usage and limitations.

### 2.1 Datasets

#### 2.1.1 CMIP6

**CMIP6.** The backbone of the ClimateSet data pipeline is the Coupled Model Intercomparison Project Phase 6 (CMIP6), an archive uniting climate model outputs from numerous sources (Eyring et al., 2016). CMIP6 is used to inform the IPCC Assessment Reports and represents the largest available archive of *comparable* climate datasets (Petrie et al., 2021; Balaji et al., 2018) with 3.7 million datasets and an expected total size of 20-80 PB. The core-data of ClimateSet are specifically climate model outputs of ScenarioMIP (O'Neill et al., 2016). ScenarioMIP contains projections of future

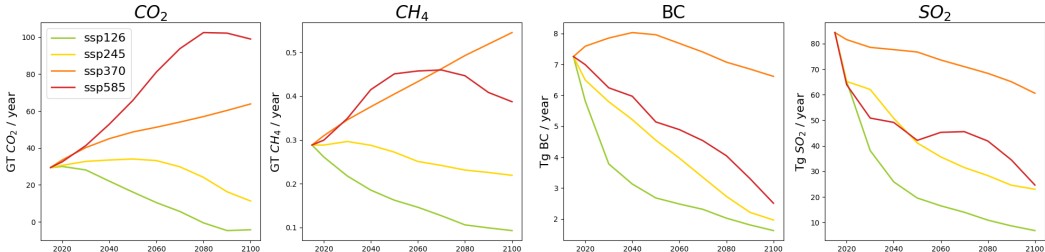

Figure 2: Forcing agent trajectories. Greenhouse gas ($CO_2$, $CH_4$), aerosol (BC), and aerosol precursor ($SO_2$) emission trajectories from 2015 – 2100 for different Shared Socio-economic pathways (SSPs).

climate change scenarios from 58 climate models[3]. Each of those climate models provides a physical simulation of how climate changes as a result of a forcing trajectory (SSP scenario) in the decades to come. The climate model receives future GHG and aerosol emission fields as input (see Section 2.1.2), and simulates outputs of climate variables such as temperature, precipitation, wind velocity, and so on. Climate models have projection uncertainties, illustrated also in Fig. 4 which shows the different temperature projections of climate models across different SSP scenarios, while Fig. 3 shows an example of different climate projections for the year 2100. Projection uncertainties arise both from (A) different climate model formulations, and (B) climate model initializations. (A) means that different climate models represent climate processes differently, leading to "inter-model variability" in the outputs. (B) means that one climate model can be initialized differently (an initialization setting is called an "ensemble member"), leading to "intra-model variability". To capture these projection uncertainties, the IPCC and policymakers rely on projections of a *set* of climate models and ensemble members. Similarly, we curated a dataset that contains the output of multiple climate models and ensemble members to reflect this projection uncertainty.

**Specifications.** For our core-dataset, we selected 36 climate models from ScenarioMIP that are summarized in Table 1. Of the 58 climate models available in ScenarioMIP we chose only those ones that had (1) *monthly frequency*, (2) at least a spatial resolution of *250 km*, (3) the scenarios *SSP1-2.6, SSP2-4.5, SSP3-7.0, and SSP5-8.5* available, resulting in a total of 36 climate models. A list of relevant features is also provided in Table 2. Other features, such as spatial resolution, grids, calendar, and units are synchronized during preprocessing (see Section 2.2). This selection ensures that ClimateSet provides the main scenarios and is spatially and temporally high enough resolved.

**Extensions.** The dataset can be extended to more climate models, ensemble members, variables, height levels, spatial, and temporal resolution, as long as the requested data is available on the Earth System Grid Federation (ESGF) server [4].

### 2.1.2 Input4MIPs

**Input4MIPs.** The Input Datasets for Model Intercomparison Projects (Input4MIPs)[5] collect the future emission trajectories of climate forcing agents that are used as input for climate models (Durack et al., 2017). Fig. C.1 in the Appendix shows an example of a GHG emission map. Similarly as climate models do, ClimateSet uses such maps as input data for its climate emulation task. We selected specifically Input4MIPs as it has been endorsed by CMIP6, i.e. it is compatible with ClimateSet's CMIP6 data, and is considered as the best climate model input data available (Durack et al., 2017). The different climate forcing trajectories included in Input4MIPs are based on different SSP scenarios, ranging from "taking the green road" (SSP1), "a rocky road" (SSP3), to "taking the highway" (SSP5). The digits after the "SSPX" term indicate the amount of radiative forcing in $W/m^2$ expected in 2100. Of the different datasets available in Input4MIPs, ClimateSet uses (1) a forcing dataset including $CO_2$, $CH_4$, $SO_2$, and Black Carbon (BC), by Feng et al. (2020), and (2) the historic open biomass burning emissions dataset by Van Marle et al. (2017).

**Specifications.** For our core-dataset, we selected four main SSP scenarios (SSP1-2.6, SSP2-4.5, SSP3-7.0, SSP5-8.5), the historical scenario, four climate forcers ($CO_2$, $CH_4$, $SO_2$, and BC). The future trajectory scenarios of those four climate forcers are represented in Fig. 2. Of the 9 scenarios

---

[3]Link to current status of ScenarioMIP data

[4]`https://esgf-node.llnl.gov/search/cmip6/`

[5]Link to current status of Input4MIPs data

| Climate Model | | | | Input4MIPs Files | |
|---|---|---|---|---|---|
| Name | Publication | Nominal resolution | Ensemble members | Historic | Future |
| ACCESS-CM2 | Bi et al. (2020) | 250 km | 5 | all-fires | all-fires |
| ACCESS-ESM1-5 | Ziehn et al. (2020) | 250 km | 40 | all-fires | all-fires |
| AWI-CM-1-1-MR | Semmler et al. (2020) | 100 km | 1 | all-fires | all-fires |
| BCC-CSM2-MR | Wu et al. (2021) | 100 km | 1 | all-fires | all-fires |
| CAMS-CSM1-0 | Hao-Ming et al. (2019) | 100 km | 2 | all-fires | all-fires |
| CAS-ESM2-0 | Zhou et al. (2020) | 100 km | 2 | all-fires | all-fires |
| CESM2 | Danabasoglu et al. (2020) | 100 km | 3 | *anthro-fires* | all-fires |
| CESM2-WACCM | Danabasoglu et al. (2020) | 100 km | 5 | *no-fires* | *no-fires* |
| CMCC-CM2-SR5 | Cherchi et al. (2019) | 100 km | 1 | all-fires | all-fires |
| CMCC-ESM2 | Lovato et al. (2022) | 100 km | 1 | *no-fires* | *no-fires* |
| CNRM-CM6-1 | Voldoire et al. (2019) | 250 km | 10 | all-fires | all-fires |
| CNRM-CM6-1-HR | Voldoire et al. (2019) | 100 km | 1 | all-fires | all-fires |
| CNRM-ESM2-1 | Séférian et al. (2019a) | 250 km | 10 | *anthro-fires* | *anthro-fires* |
| EC-Earth3 | Döscher et al. (2022a) | 100 km | 97 | all-fires | all-fires |
| EC-Earth3-Veg | Döscher et al. (2022a) | 100 km | 8 | *anthro-fires* | *anthro-fires* |
| EC-Earth3-Veg-LR | Döscher et al. (2022a) | 250 km | 3 | *anthro-fires* | *anthro-fires* |
| FGOALS-f3-L | He et al. (2019) | 100 km | 1 | all-fires | all-fires |
| FGOALS-g3 | Pu et al. (2020) | 250 km | 5 | all-fires | all-fires |
| GFDL-ESM4 | Dunne et al. (2020) | 100 km | 3 | *no-fires* | *no-fires* |
| GISS-E2-1-G | Kelley et al. (2020) | 250 km | 36 | all-fires | all-fires |
| GISS-E2-1-H | Kelley et al. (2020) | 250 km | 10 | all-fires | all-fires |
| GISS-E2-2-G | Rind et al. (2020) | 250 km | 5 | all-fires | all-fires |
| IITM-ESM | Krishnan et al. (2021) | 250 km | 1 | all-fires | all-fires |
| INM-CM4-8 | Volodin et al. (2018) | 100 km | 1 | all-fires | all-fires |
| INM-CM5-0 | Volodin and Gritsun (2018) | 100 km | 5 | all-fires | all-fires |
| IPSL-CM6A-LR | Boucher et al. (2020) | 250 km | 11 | all-fires | all-fires |
| KACE-1-0-G | Lee et al. (2020) | 250 km | 3 | all-fires | all-fires |
| MCM-UA-1-0 | Stouffer (2019) | 250 km | 1 | all-fires | all-fires |
| MIROC6 | Tatebe et al. (2019) | 250 km | 50 | all-fires | all-fires |
| MPI-ESM1-2-HR | Gutjahr et al. (2019) | 100 km | 10 | all-fires | all-fires |
| MPI-ESM1-2-LR | Mauritsen et al. (2019) | 250 km | 30 | *anthro-fires* | *anthro-fires* |
| MRI-ESM2-0 | Yukimoto et al. (2019) | 100 km | 10 | *anthro-fires* | all-fires |
| NorESM2-LM | Seland et al. (2020) | 250 km | 13 | *no-fires* | *no-fires* |
| NorESM2-MM | Seland et al. (2020) | 100 km | 2 | *no-fires* | *no-fires* |
| TaiESM1 | Wang et al. (2021) | 100 km | 1 | *anthro-fires* | all-fires |
| UKESM1-0-LL | Sellar et al. (2019) | 250 km | 17 | all-fires | all-fires |

Table 1: Climate models included in ClimateSet with related source and original nominal resolution. The ensemble members are the maximum number of ensemble members available for one scenario, i.e. one scenario does not always contain all ensemble members. Similarly, one ensemble member often contains only a subset of the scenarios. The Input4MIPs files column refers to the specific input files needed for a climate model. These input files provided by ClimateSet are representing the corresponding climate forcer emission fields, separated for different fire models (see Section 2.1.2).

| Features | CMIP6 | Input4MIPs |
|---|---|---|
| Variables | temperature, precipitation | $CO_2$, $CH_4$, BC, $SO_2$ |
| Scenarios | historical, SSP1-2.6, SSP2-4.5, SSP3-7.0, SSP5-8.5 | historical, SSP1-2.6, SSP2-4.5, SSP3-7.0, SSP5-8.5 |
| Frequency | monthly | monthly, every 10 years |
| Time length | 2015 — 2100 | 2015 — 2100 |
| Spatial area | global | global |
| Levels | 1 (surface) | 1 – 25 (AIR) |

Table 2: Features shared among the two original datasets, CMIP6 and Input4MIPs. The features are only representative for SSP scenario data. For historical data, the time length ranges from 1750 – 2015 however, some datasets only provide the subset 1850 – 2014. In terms of frequency, the historical Input4MIPs data has monthly data for *every* year. The levels for Input4MIPs data noted here refer to the different height levels of the anthropogenic AIR emission fields.

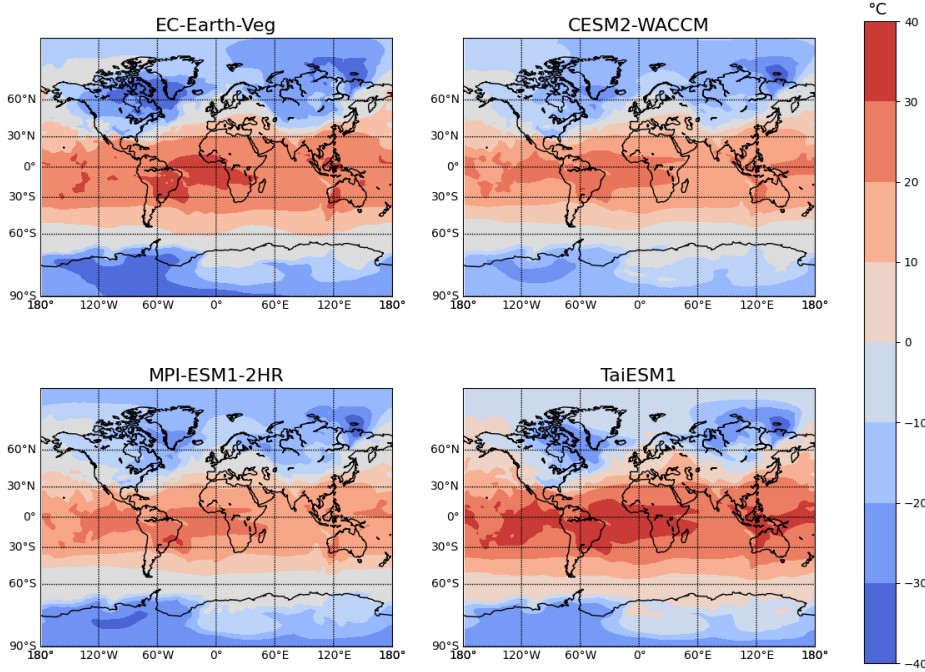

Figure 3: Climate model predictions. Absolute temperature projections for January 2100 under SSP3-7.0 by A) EC-Earth3-Veg, B) CESM2-WACCM, C) MPI-ESM1-2-HR, and D) TaiESM1.

available in Input4MIPs we have chosen the mentioned four because they are part of the five most important SSP scenarios for policy making (Arias et al., 2021). We did not include SSP1-1.9 – the lowest and most optimistic forcing scenario – because not all climate models include SSP1-1.9 and this would have narrowed ClimateSet's CMIP6 dataset significantly. The climate forcers we have chosen are the same used in Watson-Parris et al. (2022)'s ClimateBench. $CO_2$ is due to its cumulative character (see Appendix C) and high concentration considered as the most important climate forcing factor, followed by $CH_4$ with a long lifecycle and high radiative forcing potential (Fig. SPM2 in on Climate Change (IPCC) (2023)). $SO_2$ provides a cooling effect, and damages vegetation, while BC decreases the Earth's albedo when deposited and has negative effects on human health. Overall, this selection represents both long-lived GHG ($CO_2$ and $CH_4$), and short-lived aerosol and aerosol precursors ($SO_2$, BC). The Feng et al. (2020) dataset provides for each scenario and climate forcer three data-files that represent (1) "anthropogenic", (2) "anthropogenic aircraft", and (3) "open burning" emissions. Since the "open burning" emissions are not available for the historic case in Feng et al. (2020), we supplemented this data with Van Marle et al. (2017)'s dataset. ClimateSet provides all emissions in kg m$^{-2}$ s$^{-1}$. Table 2 lists the shared features of the Input4MIPs datasets. Appendix D describes how the historical open burning data should be handled and its dependence on the fire model of climate models (Appendix E). In our preprocessing pipeline the mentioned Input4MIPs datasets are combined appropriately given those considerations, i.e. ClimateSet provides summed up and ready-to-load input emission data.

**Extensions.** ClimateSet can be extended by additional scenarios (e.g. SSP1-1.9, SSP2-4.5-covid, SSP4-6.0) and climate forcers (e.g. CO, $H_2$, $NH_3$). Note, that for additional control (e.g. piControl) and CO2 scenarios (abrupt-4xCO2, 1pctCO2), no additional Input4MIPs data is required. Those scenarios are set "internally" in the climate models and can be retrieved from CMIP6. When extending the ClimateSet's input data, note that the historical data of the desired climate forcer must be available both in Feng et al. (2020)'s and Van Marle et al. (2017)'s dataset.

## 2.2 Data Collection

All data requested through ClimateSet is directly downloaded from ESGF [6] (Appendix G) and run through ClimateSet's preprocessing pipeline. The original data from ESGF is not consistent across

---

[6]https://esgf-node.llnl.gov/projects/esgf-llnl/

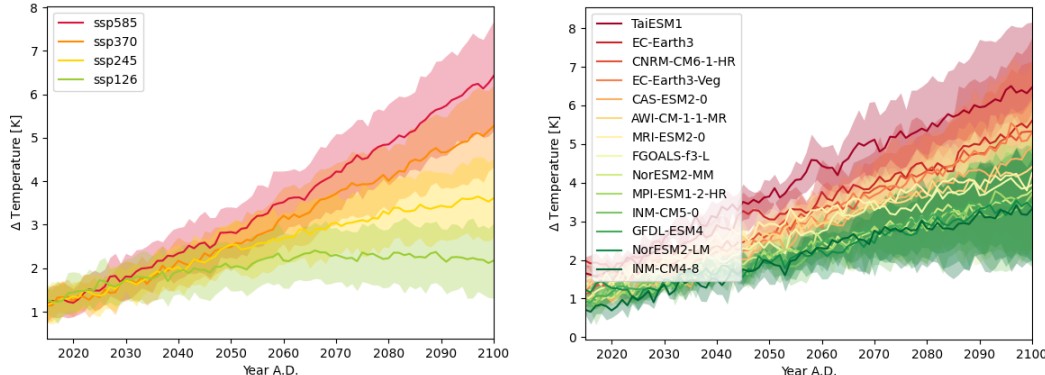

Figure 4: (A) Scenario variance across climate models, and (B) climate model variance across scenarios. The figures show the projected global temperature increase compared to the historical mean (1960 – 1990). For each scenario or model the mean temperature increase is represented as colored line and the standard deviation as area. Note the differences in temperature projections.

different datasets and climate models, and must be preprocessed. The ClimateSet preprocessor is built modularly, as described below, and can be reused and recycled for related datasets.

**The Checker** uncovers some inconsistencies across different climate models or input datasets. It checks for corruptness of files, variable naming, units, temporal and spatial resolution, and longitude-latitude structure. Based on its output, some of the preprocessing steps can be skipped if not needed.

**The Raw Processor** for CMIP6 data syncs time-axis, calendars, and height levels. For the Input4MIPs data, it additionally handles the special case of biomass burning data, corrects units, sums over sectors, and creates loadable data files. The raw processor is using Climate Data Operators (CDO) Schulzweida (2022), a command line tool optimized for processing large climate datasets. To our knowledge, this is the fastest way to process the data we have at hand (see also Appendix J).

**The Resolution Processor** creates the desired spatial and temporal resolution across all data. The spatial remapping can be used to increase or decrease the resolution. The chosen remapping algorithm can be adapted for each variable. On the temporal axis, the processor can be used to aggregate (e.g. from "months" to "years"), interpolate (e.g. from "years" to "months"), and interpolate between "time jumps" (e.g. "monthly data every 10 years" to "monthly data every year"). To make the resolution processor as efficient as possible, we implemented it with CDO. It can also be used separately from the other processors, see Appendix I for further information.

**The Structure Processor** is mainly used for CMIP6 data to make sure that all files are using the same longitude-latitude structure, vertices and bounds, the same names for variables and dimensions, and to correct units where necessary. The structure processor was implemented with xmip (Busecke and Abernathey, 2020) and follows their guidelines. Note that it is significantly slower than the CDO-implemented modules.

More details and visualizations of ClimateSet's preprocessing-pipelines can be found in Appendix H.

## 2.3 Usage

**Access ClimateSet.** Instructions to access and download the core-data can be found on `https://climateset.github.io`. We provide both the raw data and the final processed data. The latter can be directly used for climate emulation and other climate prediction related tasks.

**Extend ClimateSet.** To extend ClimateSet, you first use the downloader to retrieve the desired data from ESGF. Then, you build the desired preprocessing pipeline by adapting the configuration files or by stacking the desired modules. Every processing step can be switched on or off.

**Accelerate ClimateSet.** If run on a machine with 1 CPU core, 16GB memory, with single-threading, the complete preprocessing of the 36 climate models takes $\sim 160$ hours. The preprocessing can be accelerated in the following ways: (1) Using the multi-thread function of the CDO-implemented processors (resolution & raw); (2) waiving the checker and/or the structure processor; (3) supplying the resolution processor with an example resolution map. This is further explained in Appendix J.

## 2.4 Limitations

**Data Retrieval.** When extending the dataset, users may run into issues with the data retrieval from ESGF's server nodes. Server nodes can be down from time to time, i.e. users need to wait until those nodes are back online, which can take up to weeks (node status: `https://esgf-node.llnl.gov/status/`). Usually, only a subset of the data, e.g. one specific climate model, is affected by this.

**Computational Resources**. Depending on the size of the desired dataset, extending ClimateSet might only be feasible with access to a high-performance compute (HPC) cluster. The core dataset of ClimateSet is already relatively large with 2.0 TB of preprocessed data. Furthermore, for a set of multiple climate models, the preprocessing takes too long to be carried on a local machine that only supports single-threading for CDO (see "Accelerate ClimateSet" in Section 2.3). However, for a smaller set of climate models, storing and preprocessing the dataset on a local machine works well.

**Weighting of Climate Models**. ClimateSet currently unites 36 climate models without considering the similarity between some of them. Explanations of within- and across- climate model similarities are given in Appendix F. When training a large model on ClimateSet it would be beneficial to weight the climate models to prevent over- and under-representation of some climate models or sub-models. However, such a weighting does only exist for CMIP5 so far (Massoud et al., 2020; Wootten et al., 2020), and requires in-depth domain knowledge and is thus beyond the work presented here.

**Evaluation.** ClimateSet is limited by its current deterministic setting. Climate models are deterministic, however, uncertainties among and across them could be captured in future work. This requires weighting them as discussed above. Moreover, the evaluation metrics should be extended and adapted for different climatic variables and task settings; metrics will be updated continuously on GitHub.

**Extension beyond ScenarioMIP**. Users might want to retrieve and process data beyond ScenarioMIP. The downloader and processor pipeline of ClimateSet can be re-used to that end, however, we did not test those cases. We encourage pull and feature requests for other CMIP6-endorsed datasets.

## 3 Benchmarking Setup

**Task.** In climate emulation the objective is to simulate the output of a climate model as closely as possible. The emulator receives the same input as the climate model, but can produce climate projections for new input data a lot faster than climate models during inference time (see Fig. 5). Here, the goal is to predict a time-series of climate variables (e.g. temperature and precipitation for 2015-2100) from a given parallel time-series of climate forcer emission maps from 2015-2100. We treat the task as a diagnostic-type prediction, however, it can also be treated as an autoregressive task. Refer to Watson-Parris et al. (2022) for further explanations about emulation. We use two different versions of climate emulators: (1) Single-Emulators, and (2) Super-Emulators. By "Single-Emulator" we refer to an emulator that is trained on a *single* climate model. By "Super-Emulator" we refer to an emulator that is trained on a *set* of climate models, and is able to project the climate responses of all the participating climate models. The super-emulator is explained in more detail in Appendix L.2.

**ML Models.** We trained most types of models that have been used for climate emulation on ClimateBench (Watson-Parris et al., 2022) to date: Convolutional long short-term memory (ConvLSTM) (Hochreiter and Schmidhuber, 1997; LeCun et al., 1989; Watson-Parris et al., 2022), Gaussian Process regression (GP) (Williams and Rasmussen, 2006; Hensman et al., 2015), and ClimaX (Nguyen et al., 2023). ClimaX is the current state-of-the-art ML model on ClimateBench. We omitted the Random Forest (RF) (Breiman, 2001) since we could not fully reproduce ClimateBench's experiments with our training configuration of predicting two variables concurrently. We added a U-Net (Ronneberger et al., 2015) as a simple baseline. Where necessary, we adapted ClimateBench's implementations. The implementation details and differences to the original models are described in Appendix K.

**Data.** Each emulator receives as input the climate forcing emission fields ($CO_2$, $CH_4$, $SO_2$, BC), and as target the output variables of climate models (temperature, precipitation). All data was processed to have a spatial resolution of approximately 250 km (144 x 96 longitude-latitude cells) and a temporal resolution of monthly data. For both the input and target data, there are 86-year time-series available for the 4 SSP scenarios (2015 – 2100), and 165 years for the historical scenario (1850 – 2014). Those time-series are divided into 1-year chunks. The resulting data has a shape of (`scenarios * years * months, variables, longitude, latitude`). Assuming we choose 86 years and four climate forcers, we would map from input shape (5*86*12, 4, 144, 96) to output (5*86*12, 2, 144, 96).

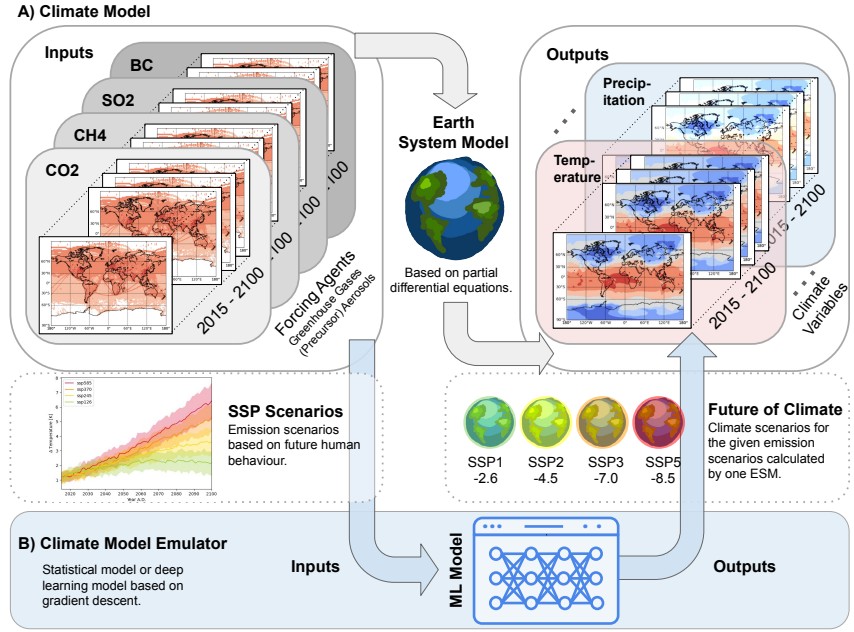

Figure 5: Climate emulation. A) A climate model receives the future trajectories of climate forcers as input, and uses this input to simulate the future climate with the help of its different components (atmosphere, ocean, ice, land, and fire models). It outputs how the climate (e.g. temperature) would respond to a given emission scenario. B) A climate model emulator receives the same inputs as a climate model and learns to emulate its outputs. After training on several scenarios and ensemble members, it can predict new scenarios much faster than a traditional climate model.

**Train-Test-Split**. For training and validation, the historical scenario, SSP1-2.6, SSP3-7.0, and SSP5-8.5 are used. A random 10% split of the data is hold out for validation, and SSP2.45 for testing.

**Experiments.** We run experiments on (A) single-emulation, (B) super-emulation, and (C) generalization capabilities of the different ML models. For single-emulation, each ML model is trained on each of the 15 climate models separately (i.e., we result with 15 independent ML models). Our models are trained on our internal cluster, using a single Nvidia-RTX8000 with 32GB of RAM. For the super-emulation task, a single ML model is trained on 6 climate models together to demonstrate how super-emulation works (Appendix L.2). Future work can extend the super-emulator to run on all 36 climate models. During inference, the super-emulator can predict novel scenarios for each climate model that participated in training.

**Further Notes.** To test the generalization capabilities of the single-emulator, we use its weights, train on a climate model, finetune on NorESM2-LM and compared the results with a single-emulator trained only on NorESM2-LM. All experiments included only one ensemble member; future experiments could evaluate the influence of intra-model variability on ML model performance further. For all experiments we used the latitude-longitude weighted root mean squared error (RMSE) as implemented in (Nguyen et al., 2023) as main evaluation metric. Refer to Appendix L for additional details.

## 4 Benchmarking Results

Here, we present a subset of our climate emulation results to investigate the differences between using a dataset that contains a single climate model, and one that contains multiple climate models. Fig. 6 shows the RMSE of temperature projections for the test scenario (SSP2-4.5) among the neural-network based models (ClimaX, ClimaX Frozen, ConvLSTM, and U-Net) on a subset of six climate models (NorESM2-LM, NorESM2-MM, MPI-ESM1-2-HR, GFDL-ESM4, TaiESM1, and EC-Earth3). With the exception of the U-Net, the named ML models had been considered in (Nguyen et al., 2023) on NorESM2-LM data retrieved from ClimateBench. We find the performance of all ML models on NorESM2-LM to be in a similar range as reported in (Nguyen et al., 2023). ClimaX, ClimateBench, and ClimateSet observed all different performance values for ConvLSTM on NorESM2-LM due to differences in the implementations, e.g. ClimateSet's ConvLSTM outperformed ClimaX's ConvLSTM version (0.3 vs. 0.4). In general, the RMSE values are in line with previous

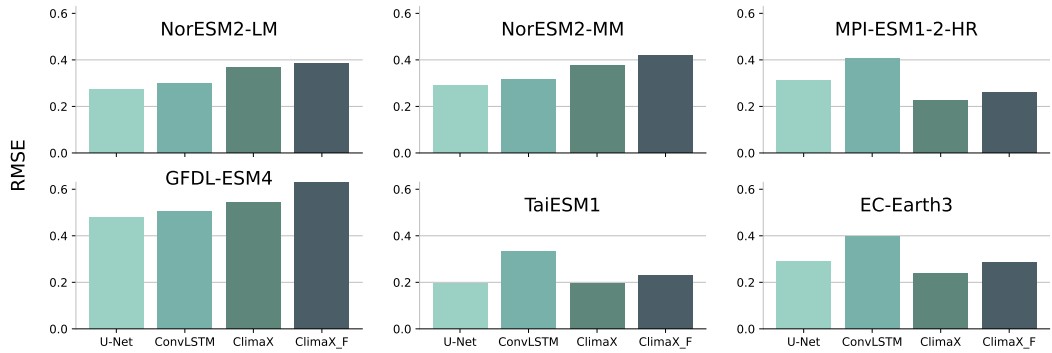

Figure 6: Emulation benchmark. RMSE for the temperature projection of SSP2-4.5 (2015 – 2100) for the ML models ClimaX Frozen, ClimaX, ConvLSTM, and U-Net. Each ML model (color coded) was trained separately on the climate models NorESM2-LM, NorESM2-MM, MPI-ESM1-2-HR, GFDL-ESM4, TaiESM1, and EC-Earth3. The RMSE values (lower is better) are latitude-longitude weighted and averaged over three seeds for each ML model.

work where comparable. Notably, simple models such as the U-Net and ConvLSTM (we applied no significant tuning or adaptions to those models) can keep up with ClimaX. U-Net even outperformed ClimaX consistently when dropping ClimaX's warm-up period. Refer to Appendix M.1 for more single-emulator results. This shows that for climate emulation tasks significant performance gains can still be made with relatively simple baselines that had not yet been investigated for this task.

While the different ML models showed consistent performance across different climate models, there were exceptions. For example, when looking only at the NorESM2 models, it seems that ClimateSet's ConvLSTM significantly outperforms both ClimaX and ClimaX Frozen (Fig. 6). When looking across several climate models, however, it becomes quickly apparent that both ClimaX and ClimaX Frozen outperform the ConvLSTM across multiple climate models. The consistency of the performance might depend on (A) whether an ML model overfits one climate model (i.e. the hyperparameters match this model in particular), and (B) whether an ML model is particularly good at generalizing across different climate models. Refer to Appendix M.2 and Table 5 to see the ML models performance across multiple climate models. Figure 6 shows also the climate model on which the best performance could be achieved (TaiESM1) and the worst (GFDL-ESM4). Those climate models were the best/worst across all ML models, indicating that some climate models are easier to emulate than others. In summary, the results show that testing and comparing ML models on just one climate model is not sufficient for finding the "best" ML emulator; instead, ML models should be evaluated on a set of climate models - posing different levels of difficulty - to draw such conclusions.

When training ML-models on super-emulation, on six climate models, the leader-board of the ML-models inverts (Appendix M.2): The simplest model (ConvLSTM) yields the best, and the most complex model (ClimaX) the worst performance. The experiments to test generalizability (Appendix M.3) could explain this: The simpler models are better at generalizing across several climate models, while the more complex models fail to generalize as well. Future work is needed to investigate if larger ML models simply need a longer training period to learn a full, generalizable, representation of climate models. The results show that ClimateSet can be used to identify ML models generalizing across different climate models. Thus, ClimateSet can be the basis for developing new climate emulators that can emulate climate models across the CMIP6 archive instead of only a single one.

## 5 Conclusion

With ClimateSet, we respond to the widely expressed need for a large-scale, consistent climate model dataset for machine learning. Among other tasks, ClimateSet can be used to reveal new insights for ML climate emulation. We found that ClimaX is the best single-emulator, while ConvLSTM generalizes better and is the best super-emulator across multiple climate models. We found the overall ranking of ML models varied across different climate model datasets, showing the need for ClimateSet as a unified benchmark. ClimateSet also provides a pipeline to retrieve and preprocess further climate model data consistently. We envision this will be particularly useful for researchers who need large training datasets, e.g. for climate foundation models. We hope ClimateSet enables the ML-community to address a much wider range of climate-related tasks. Tackling those tasks on the scale of CMIP6 will help the ML-community to contribute meaningfully to climate policy making.

## Acknowledgments and Disclosure of Funding

We acknowledge the WCRP, which, through its Working Group on Coupled Modeling, coordinated and promoted CMIP6. We thank the climate modeling groups for producing and making available their model output, the Earth System Grid Federation (ESGF) for archiving the data and providing access and the multiple funding agencies that support CMIP6 and ESGF. We would like to thank Sonia Yousfi for her helpful suggestions, Trevor Greco for editing guidance, and Sébastien Lachapelle and Julien Boussard for insightful discussions. Many thanks go to Björn Lütjens for being ClimateSet's beta tester and helping us to improve it with his valuable and deliberate feedback. Special thanks go to Boris Sakschewski for lending us his wonderful 3d graphic work on ESMs, and Steven J. Smith for his supportive explanations on the Input4MIPs data.

This project was supported by the Intel-Mila partnership program and Canada CIFAR AI Chairs program. J.R. has received funding from the European Research Council (ERC) Starting Grant CausalEarth under the European Union's Horizon 2020 research and innovation program (Grant Agreement No. 948112). P.N. was supported by UK Natural Environment Research Council (grant no. NE/V012045/1).

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

## A  Data and Code Availability

Access to code and data is provided on `https://climateset.github.io`.

## B  Overview of Related ML-Datasets

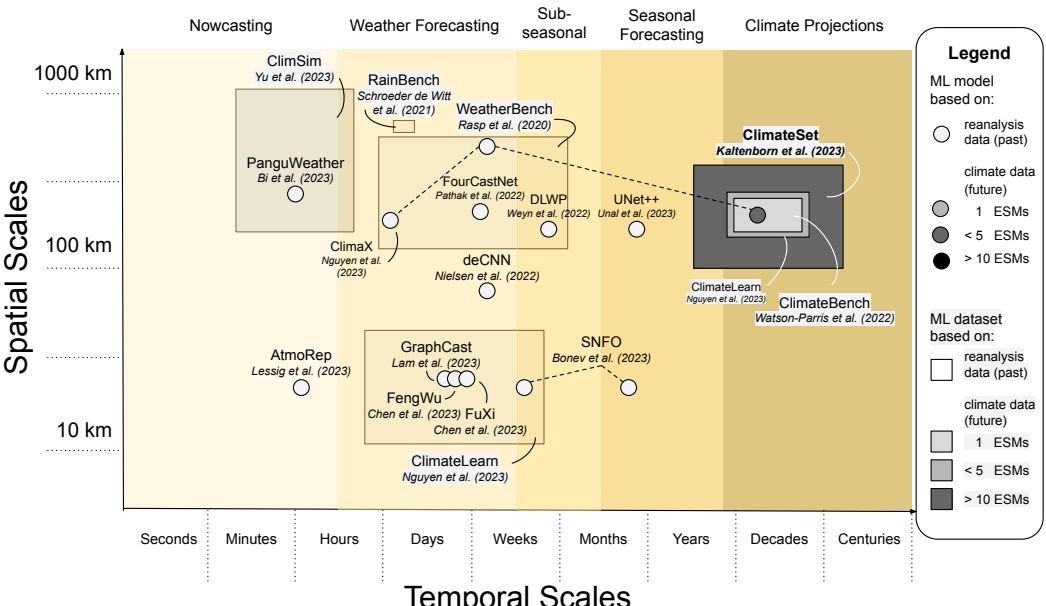

Figure 7: ML datasets and models used in the weather and climate domain. Only models and datasets are included that operate on a global scale and use either re-analysis or climate model data. The dots represent ML-models, the squares datasets. The area covered by the datasets shows to which spatial and temporal scales it extends. The models and datasetes are colored darker when several climate models (ESMs) are used. Some ML-models were run for several tasks on different temporal and spatial scales; their tasks are represented by different points that are connected with dotted lines. The figure illustrates that related ML work has mostly focused on the weather scale and not the centurial climate scale. It also shows that high number of climate models has been under-utilized in the ML community so far. The figure was inspired by Mukkavilli et al. (2023).

Datasets:

- ClimSim (Yu et al., 2023)
- RainBench (de Witt et al., 2021)
- WeatherBench (Rasp et al., 2020) and WeatherBench2.0 (Rasp et al., 2023)
- ClimateLearn (Nguyen et al., 2023)
- ClimateBench (Watson-Parris et al., 2022)

Models:

- PanguWeather (Bi et al., 2022)
- AtmoRep (Lessig et al., 2023)
- ClimaX (Nguyen et al., 2023)
- FourCastNet (Pathak et al., 2022)
- deCNN (Nielsen et al., 2022)
- GraphCast (Lam et al., 2022)

- FengWu (Chen et al., 2023)
- FuXi (Chen et al., 2023)
- DLWP (Weyn et al., 2021)
- SNFO (Bonev et al., 2023)
- UNet++ (Unal et al., 2023)

## C    CO2 Data

The greenhouse gas CO2 requires special handling within this dataset and within climate emulation in particular. In Fig. C.1, an example of a CO2 emission map is shown.

One reason for the special handling is the accumulative nature of CO2 (Canadell et al., 2007), i.e. it does not have a half-life as other greenhouse gases or aerosols do. While the carbon cycle does have natural sinks and sources, the anthropogenic CO2 emissions are still considered accumulative, since the CO2 does get added to the overall carbon-cycle instead of being broken down as in the case of other GHG gases. Consequently, we oversimplify and loose important information if we only use short time-chunks of CO2 emissions. The following two approaches could be used to address this problem: 1) Using the full time-series of CO2 emissions, i.e. mapping 85 years of GHG emissions to 85 years of climate response. 2) Using the cumulative CO2 emissions. Presently, our dataset does not provide the cumulative CO2 emissions for the user to select between options (1) and (2), however, we might add the cumulative data in the future.

The other reason why CO2 requires special handling is that the "fire datasets" (historical biomass-burning and future openburning) Van Marle et al. (2017); Feng et al. (2020) do not contain CO2. The historic biomassburning data simply does not have this information to date. The future openburning dataset might not have this data because it partially builds upon the historic biomassburning dataset Van Marle et al. (2017). This leads to the the file-structure of our dataset containing "all-fires", "anthro-fires", "no-fires" for all greenhouse gases except CO2. CO2 has only one type of file ("sum"), summing up the anthropogenic emissions and the aircraft emissions. This is true for both the historical and the SSP data. We will update our dataset in case future publications targeting Input4MIPs provide openburning and biomassburning data.

Another case where CO2 data needs additional preprocessing is when control scenarios rather than SSP scenarios are included in the dataset. Example for control scenarios are abrupt-4xCO2 (abrupt quadrupling of CO2 in the atmosphere), piControl (pre-industrial CO2 kept constant). Those scenarios are especially interesting when interventional data is needed since one variable is changed (intervened on) while the others stay constant. For those scenarios, the internal CO2 mass balances of the climate models must be used: The abrupt-4xCO2 scenario is the scenario of quadrupling the current climate-model-internal CO2 amount. The CO2 mass balances can be downloaded for each climate model from Earth System Grid Federation system (ESGF).

### C.1    CO2 Emission Map

An example of a CO2 emission map is shown in Fig. 8.

## D    Open Burning and Biomass Burning Data

The emission datasets usually contain three different types of data: 1) Anthropogenic emissions, 2) Aircraft emissions, and 3) "fire" emissions. This data is collected across two different datasets, one for the historical emissions (Van Marle et al., 2017) and one for the future emissions (Feng et al., 2020). Due to different naming conventions in Van Marle et al. (2017) and Feng et al. (2020), the "fire emissions" are called "openburning emissions" for the future case, and "biomassburning emissions" in the past case. We use both names here to distinguish between the two datasets, because – in contrast to the anthropogenic and aircraft emissions – the openburning and biomassburning data need different types of treatment.

All three data types (1-3) have a spatial and temporal dimension, and all three of them separate the emissions on another dimension: (1) The anthropogenic emission files contain different sectors, (2) the aircraft emissions files have different height levels, and (3) the "fire" emissions consist of different types of fires. However, in contrast to (1) and (2), the "fire" emissions in the datasets contain only *one* level, i.e. the different fire types are not directly encoded in the datasets. While the sectors and levels of the anthropogenic and the aircraft emission files are simply summed up, we actually need to separate between the different types of "fire emissions" depending on the fire model of each climate model (see Appendix E).

To retrieve individual numbers for the different types of fire emissions, the percentage file that are provided alongside with the original datasets can be used. Those percentage files are available on the

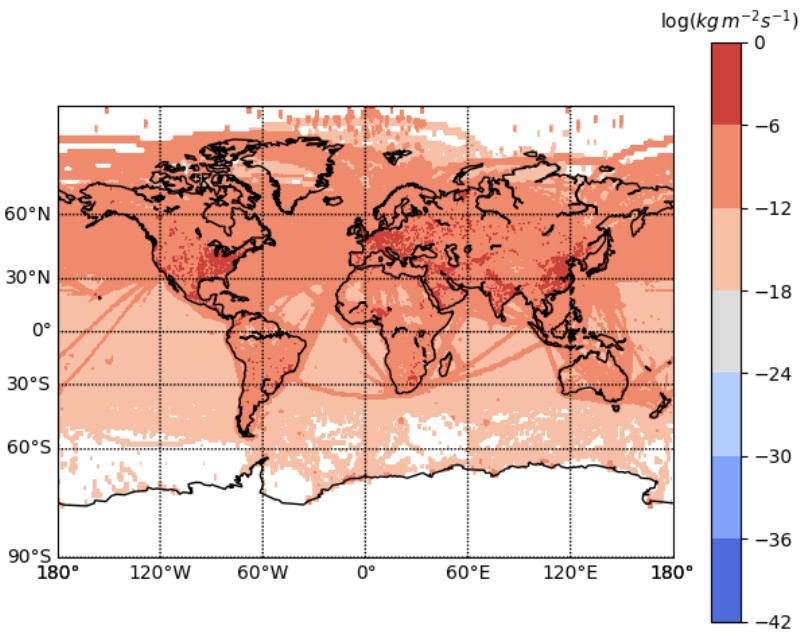

Figure 8: Input4MIP's projected global anthropogenic and aircraft CO2 emissions for January 2100 in the SSP3-7.0. The emissions are summed over all sectors. Transportation paths become visible in the CO2 emission map as well as CO2 emissions across land masses.

Input4MIPs archive on ESGF. The following sectors exist for those files, and are different for the historical and future data:

Biomassburning:

- Savanna, grassland, and shrubland fires
- Boreal forest fires
- Temperate forest fires
- Deforestation and degradation
- Peatland fires
- Agricultural waste burning

Openburning:

- Agricultural waste burning on fields
- Forest burning
- Grassland burning
- Peat burning

The preprocessing pipeline provided here computes the overall fire emission maps and separates between three final cases: (1) anthropogenic fire emissions only, (2) all fire emission sources, (3) no fire emissions at all. Note that for CO2 only one type exists (no "fire" emissions), as further described in Appendix E. Which of those three files should be used in the other GHG cases, depends on the fire model of the climate model and is listed in Table 3. The subsequent section analyses in detail which types need to be included for each climate models.

# E  Fire Models

The internal fire model of each climate model determines which GHG emission fields (Lasslop et al., 2020) should be used as input for the climate model. For example, if a climate model contains an extensive fire model that already models peatland burning, we cannot provide peatland burning emissions as input to the model – the emissions would be "counted" twice. Hence, it is necessary to evaluate for each climate model which fire model it is using, what the model is capable of, and which fire emissions should consequently be provided as input. Table 3 provides an overview of all the models, which fire model they are using and which types of fires need to be included in their input data. In the preprocessing pipeline this data is summarized to anthropogenic fire emissions ("anthro-fires"), all fire emissions ("all-fires"), and no fire emissions ("no-fires"). The latter can be used if a fire model is capable of modeling all fires such as the NorESM2. Matching the right GHG emission inputs to the climate models output is definitely good practice, however, we want to mention that in the single-emulator case, ML models are expected to be adequate for mapping from simple emission files (i.e. ignoring fire emissions) to their climatic response.

|  | Model | Land Model | Historical fire emission types | Future emission types | Type of fire model |
|---|---|---|---|---|---|
| No fire model | All other ESMs | None | All | All | No fire model |
| Historic + future fire model | CESM2-WACCM | CLM5 | None | None | w/ anthro. |
|  | CNRM-ESM2-1 | SURFEXv8.0 (ISBA) | Deforestation + Agricultural | Deforestation + Agriculture | w/o anthro. |
|  | CMCC-ESM2 | CLM4.5 | None | None | w/ anthro. |
|  | EC-Earth3-Veg | LPJ-GUESSv4 | Deforestation + Agricultural | Deforestation + Agriculture | w/o anthro. |
|  | EC-Earth3-Veg-LR | LPJ-GUESSv4 | Deforestation + Agricultural | Deforestation + Agriculture | w/o anthro. |
|  | MPI-ESM1-2-LR | JSBACH3.20 | Deforestation + Agricultural | Deforestation + Agriculture | w/o anthro. |
|  | NorESM2-LM | CLM5 | None | None | w/ anthro. |
|  | NorESM2-MM | CLM5 | None | None | w/ anthro. |
|  | GFDL-ESM4 | LM4.1 | None | None | w/ anthro. |
| Historic fire model | TaiESM1 | CLM4 | Deforestation + Agricultural | All | w/o anthro. |
|  | CESM2 | CLM5 | None | All | w/ anthro. |
|  | MRI-ESM (2.0) | HAL1.0 | Deforestation + Agricultural | All | w/o anthro. |

Table 3: Fire Models with their corresponding Historical and Future emissions type. The type of fire model used by each climate model is listed in the last column. The historic fire models must use the "all" fire emission types and future emission types.

We retrieved the information about the fire models from a combination of different sources. Yu et al. (2022) describes the fire models of most the fire models we are using, while the TaiESM stems from Lasslop et al. (2020). However, the information provided there is not explicit about which fire types exactly must be include from the openburning / biomassburning files. Teckentrup et al. (2019); Rabin et al. (2017) explicitly state how different fire models treat cropland, pasture, and deforestation fire. Additionally, we referred to Séférian et al. (2019b) for CNRM-ESM2-1, Ward et al. (2018) for GFDL-ESM4, Yukimoto et al. (2019) for MRI-ESM2, Lindeskog et al. (2013) and Spessa et al. (2013) and Döscher et al. (2022b) for the EC-Earth3-Veg models. We could not find any information about the land (HAL 1.0) or fire model of the MRI-ESM2 model. However, in the meta information of their data (e.g. on esgf), they describe the "Carbon Mass Flux into Atmosphere Due to CO2 emissions from Fire Excluding Land-Use Change [kgC m-2 s-1]". From that we infer that no anthropogenic fire is modelled here. A detailed report on how we compiled the fire model information is available on request.

# F  Similarities Within and Across Climate Models

**Similarities within climate models**. When training on multiple – instead of single – ensemble member of climate models, weighting between the different ensemble members must be considered: Some models such as EC-Earth3-Veg contain up to 97 ensemble members while others contain only 1 ensemble member. Training an ML model with the complete dataset without weighting the ensemble members would skew the results heavily towards those climate models with many ensemble members. Furthermore, some ensemble members are closer to each other than others and providing this information to the ML models could potentially improve their performance.

**Similarities across climate models**. The following similarities can occur:

(A) Climate models from different institutes share the same sub-models
(e.g. ACCESS-CM2 uses the same atmospheric model as HadGEM3 models);

(B) Climate models are newer versions of themselves, stemming from the same institute
(e.g. NorESM1 (CMIP5) and NorESM2 (CMIP6));

(C) Climate models are implemented for different resolutions, with differing physics implementations to ensure resolving the relevant processes
(e.g. NorESM2-LM (100 km) and NorESM2-MM (250 km)).

Similar climate model names (e.g. GISS-E2-1-G, GISS-E2-1-H, GISS-E2-2-G) usually indicate that the same model is run with different sub-models for atmosphere, ocean, land, etc. (case (A)).

In future work, we would like to encode similarities within and across climate models in the super-emulator, for example by providing an additional input vector encoding the submodels of the climate model.

# G    Downloader

We present a ready-to use downloader class that forms the first step of creating a custom climate data dataset in an intuitive and script-based fashion.

The Downloader class can be prompted with a set of properties / heuristic to select them (for an example, see Fig. 9. Acting on those, the Downloader will automatically interact with the ESGF nodes to narrow down the search space and obtain the data if available.

Variables and experiments have to be fixed for all data. The version can be nailed down to a specific version e.g. '20170519' or alternatively, can be set to 'latest' in which case always the latest available version of the specified data files will be selected for download. For data coming from the CMIP6 project, the model ID has to be set additionally. For CMIP6, either all available ensemble members can be considered, alternatively, the user can the ensemble member ID or restrict the number of members to be considered.

The resulting search space will then be searched and constraint for the remaining properties. If the specified default nominal resolution, frequency or grid label are not available, the Downloader will download the first available alternative.

The Downloader communicates with the data nodes of the Earth System Grid Federation system (ESGF) via ESGF PyClient.

## G.1    Earth System Grid Federation (ESGF)

The Earth System Grid Federation (ESGF) is a partnership of climate modelling centres dedicated to supporting climate research by making an effort to provide access to the distributed climate model data collected by the CMIP projects, which exceeds hundreds of petabytes. The data, hosted on several servers all around the world, can be searched over a web-based platform.

Further documentation on ESGF's web presence can be found here: ESGF User Support

## G.2    ESGF Pyclient

As the ESGF's web presence proves to be quite slow and inflexible when it comes to broadly searching the database in order to craft custom datasets, and more overly cannot be easily incorporated in an automatic pipeline as it is dependent on manual selection, we resolve to make use of a python package that hooks onto the search nodes and interfaces with the ESGF Search API. This open-source package, named ESGF-Pyclient Documentation, allows querying the database via python scripting.

## G.3    Data Structure

The Downloader will download and store the raw data files separately for each unique pair of specifications, whereas the resulting files span over one year each, with the number of data points per file determined by the chosen temporal resolutions. See Fig. 10 for an illustration of the folder

```
class Downloader:
    """
    Class handling the downloading of the data. It communicates with
                                    the esgf nodes to search and
                                    download the specified data.
    """

    def __init__(
        self,
        model: str = "NorESM2-LM",   # default as in ClimateBench
        experiments: List[str] = [
            "historical",
            "ssp370",
            "hist-GHG",
            "piControl",
            "ssp434",
            "ssp126",
        ],   # sub-selection of ClimateBench default
        vars: List[str] = ["tas", "pr", "SO2", "BC"],
        data_dir: str = "../../tmp/data/",
        max_ensemble_members: int = 10, #max ensemble members
        ensemble_members: List[str] = None #preferred ensemble members
                                    used, if None not
                                    considered
    ):
```

```
downloader = Downloader(**kwargs)
downloader.download_from_model()
downloader.download_raw_input()
```

Figure 9: **The Downloader class:** The Downloader can be instantiated with the default arguments or be prompted with a new set of specifications. Downloading CMIP data from a model and downloading input4mips data are handled by separate functions, each of which can also be prompted with a new set of specifications to allow for flexibility.

structure that will be created in the data directory pointed to by the '$data\_dir$' variable that the Downloader was initialized with.

# H    Processing Pipelines

In the following, the specific pipelines are described that were used here to create the core dataset of ClimateSet. The different modules of the processing pipelines can be shuffled around and adapted as needed. The three main modules are:

- **The checker** (checks for corruptness, inconsistencies in units, variable namings, and similar);
- **The resolution processor** (spatial and temporal aggregations and interpolations, see Appendix I)
- **The raw processer** (corrects names, units, calendars, time axis, etc.)

The modules are partially different for the Input4MIPs and CMIP6 dataset, e.g. the raw processor of the CMIP6 dataset contains both an internal processor and a xmip Busecke and Abernathey (2020) processor, whereas the Input4MIPs pipeline can only use the processing tools provided by us so far. The details are described further in Fig. 11 and Fig. 12.

## H.1    Input4MIPs Processing Pipeline

The flowchart of the processing pipeline is as shown in Fig.11.

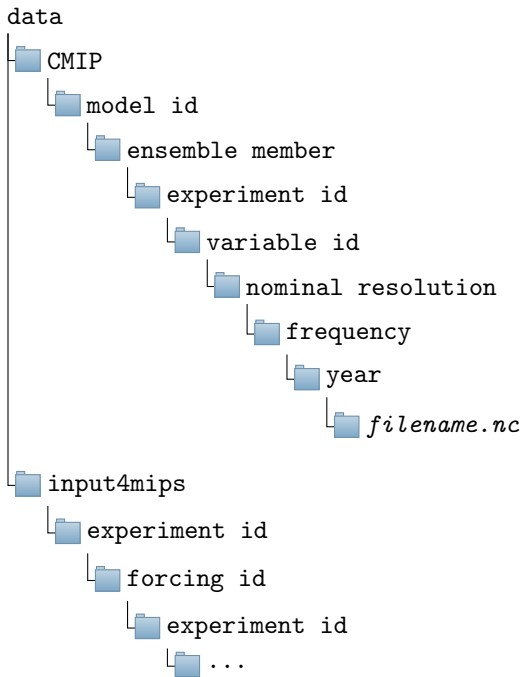

Figure 10: **Raw Data Structure:** Raw data will be stored separately for each unique pair of specifications, with one file per year.

## H.2 CMIP6 Processing Pipeline

The flowchart of the processing pipeline is as shown in Fig. 12.

# I Resolution Preprocessing

The resolution preprocessing covers both the spatial and the temporal domain, and the user can both interpolate or aggregate the data at hand. The resolution preprocessor was implemented with CDO (Schulzweida, 2022) since - to our knowledge - it is the fastest tool for remapping netcdf files. Both the resolution preprocessor and the raw preprocessor can also be extended with "National Center of Atmospheric Research Command Language" (NCL by NCAR (2019)) commands if CDO does not provide a desired functionality since NCL is more expressive than CDO. Python tools require loading the data, e.g. with xarray which takes a considerable amount of time, and CDO profits from multiprocessing and pre-weight calculations. In the following the different types of resolution processing are described that we implemented in ClimateSet with CDO. Our python code directly calls the CDO commands via "subprocess". The resolution processer can also be used separately from the rest of ClimateSet if considered helpful.

## I.1 Spatial Resolution Preprocessing

The spatial resolution preprocessor uses the CDO command "remap" which can be used for both aggregation and interpolation. Essentially, it regrids a given netcdf file. The file can be either remapped according to a given example (e.g. the NorESM grid), or according to a target gridfile. Please refer to our GitHub repo for an example of a grid or to the CDO documentation (Schulzweida, 2022). Usually, such a file contains the variables "gridtype", "xsize", "ysize", "xfirst", "yfirst", "xinc", and "yinc". If an example file is given, the spatial resolution preprocessor will pick a random file from the directory that should be processed, assuming that all files in that directory have the same grid. It will then calculate weights for this file, and the weights can then be re-used for the rest of the directory. This accelerates the resolution processing a lot. The "remap" function of CDO works bidirectional, i.e. both aggregation and interpolation is handled this way. Beware, that also climate models with the right resolution, still need to be remapped since many models have different grids

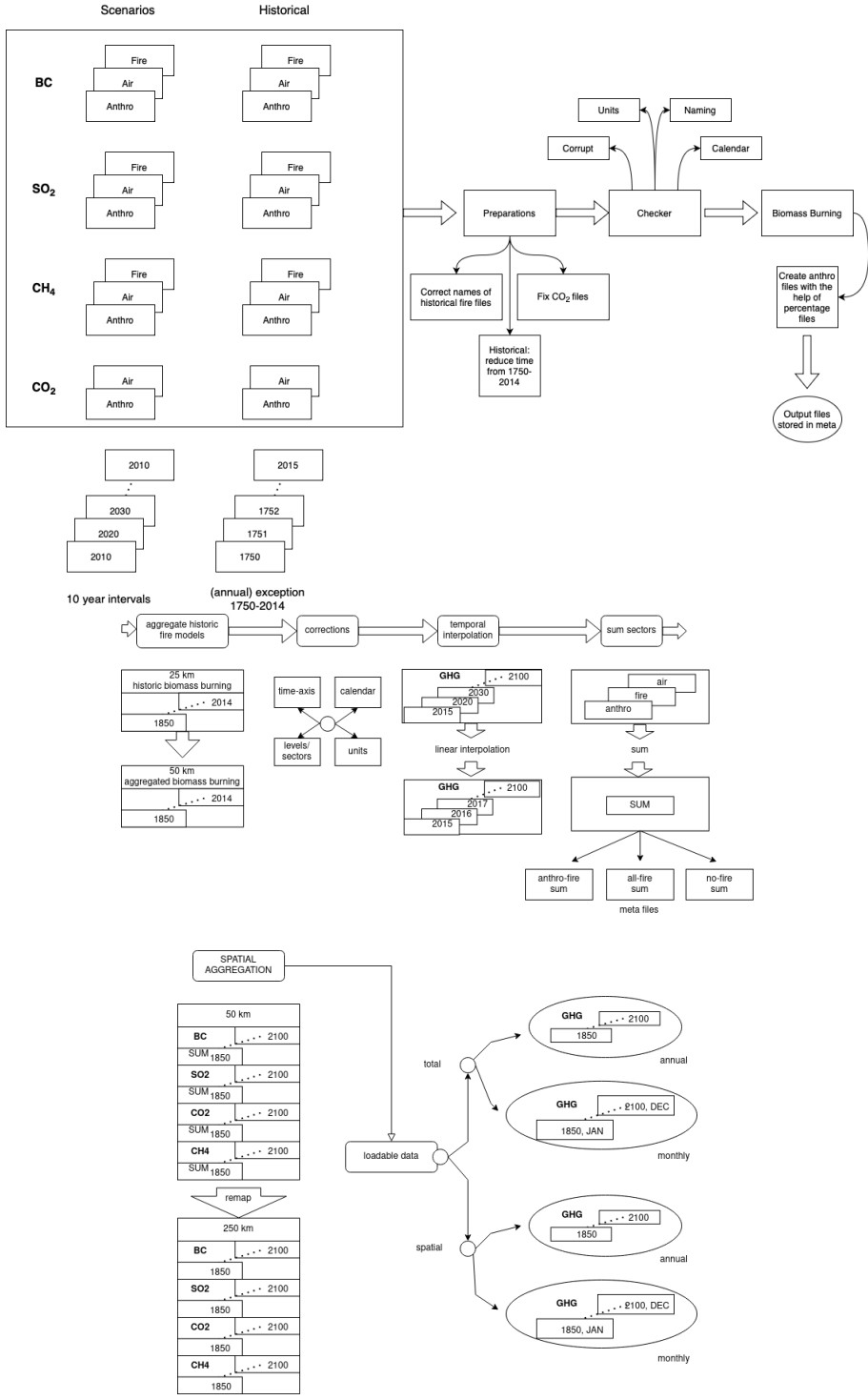

Figure 11: Flowchart of the Input4MIPs processing pipeline. The input is first repaired ($CO_2$ files, $CH_4$ files, correct biomassburning names). Subsequently, the data is checked. Anthropogenic fire emission files are created. Historical biomassburning files are aggregated to 50 km. Raw preprocessing is applied (correcting calendar, time-axis, and units, and summing levels/sectors). For the future files a temporal interpolation is applied to retrieve annual files (instead of decadal files). The different emission files are summed up (anthropogenic, aircraft, fire emissions) - with three options (anthropogenic fires, all fires, no fires). All emission maps are then aggregated to 250 km (same as climate model's resolutions). In the end loadable data is created, with a choice of receiving a spatial map, global totals, monthly or annual data.

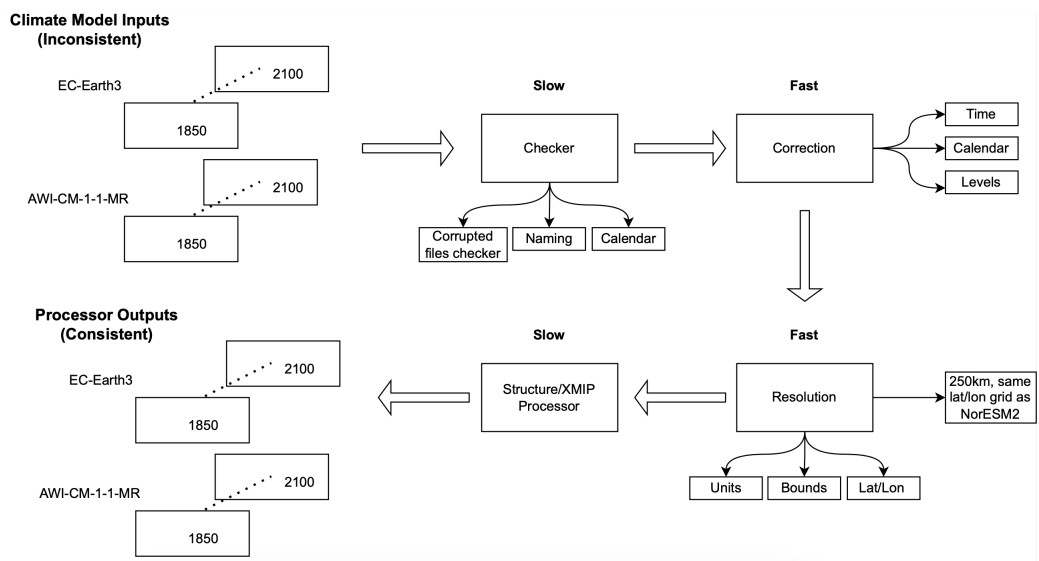

Figure 12: Flowchart of the CMIP6 processing pipeline. The checker evaluates if files are corrupt, units, names, or similar are inconsistent. The "corrections module" makes sure time-axis, calendars, and levels are aligned across all models. Afterwards, the model is aggregated to 250 km or the grid is adapted to 250 km grid of the NorESM2 model. Lastly, the xmip processer is applied to get the same structured, numpy-friendly, format across all climate models.

and different version of the same resolution. Remapping is necessary for all climate models to make sure that they are operating on the same grid.

Different algorithms can be used to interpolate to a different resolution. Those algorithms can be set in "res_processing_params.json". For example, we choose bilinear interpolation for temperature files, and first order conservative interpolation for precipitation files.

### I.2    Temporal Resolution Preprocessing

**Temporal Aggregation** A given file can be aggregated along the temporal axis to a desired time unit. So far the units "hour", "day", "month", and "year" are supported. It requires that all timesteps are set to the first of the original time unit. Alternatively, the aggregation can happen over a given number of timesteps.

**Temporal Interpolation** The temporal interpolation can have two different forms. 1) Files can be interpolated that have e.g. monthly data, but only every 10 years. In that case the "interpolate_jump" function interpolates bilinearly the missing years. 2) Files can be interpolated to different time units. This is currently supported for "seconds", "minutes", "hours", "days", "months", and "years". The interpolation can either happen to another unit or by choosing how many timesteps should be interpolated (i.e. how many new timesteps should be created between two given timesteps). All temporal interpolation is linear. For further information refer to CDOs documentation (`https://code.mpimet.mpg.de/projects/cdo/embedded/cdo.pdf`).

## J    Accelerate ClimateSet

The following measures can be applied to accelerate ClimateSet's preprocessing:

1. Using the multi-thread function of the CDO-implemented processors (resolution & raw preprocessor);
2. Waiving the checker and/or the structure processor;
3. Supplying the resolution processor with an example resolution map.

(1) only works on machines that allow multi-threading for CDO, e.g. on HPCs.

(2) is accelerating the process because the checker and the structure processor (xmip)(Busecke and Abernathey, 2020) are not implemented with CDO. Since python tools such as xmip(Busecke and Abernathey, 2020) are computationally not as optimized as CDO is, those modules are significantly slower. While the checker can be waived, the structural preprocessor can only be waived if the selected climate models use the same naming and structure across their dimensions. We would like to implement multi-threading for the checker and structure processor later on, so those two processors are not a bottleneck for the overall acceleration of the preprocessing.

(3) Is already the default. For ClimateSet a 250 km NorESM2-LM spatial map is used which can be found in the "meta" directory of the dataset. Using an example map accelerates the process by re-using the weights needed for remapping: Each file with the same longitude-latitude grid can use the same weights file for computing the new grid.

# K    ML Models' Implementation Details

## K.1    ClimaX

ClimaX stands for a foundation model that has recently emerged for tackling a wide range of weather and climate modelling tasks. With its remarkable capability to handle diverse spatial and temporal resolutions, as well as high-dimensional data, ClimaX sets itself apart by offering forecasting abilities for various lead-times and settings. It has not only achieved state-of-the-art (SOTA) or near-SOTA performance but has also demonstrated computational feasibility across multiple forecasting and climate modelling domains, including ClimateBench (Nguyen et al., 2023).

One of ClimaX's key attributes is its capacity to train on heterogeneous datasets encompassing different variables with distinct physical grounding and spatio-temporal coverage. As a foundation model, ClimaX undergoes a pre-training phase on CMIP6 data, employing a self-supervised approach. The pre-training process incorporates historical projections drawn from five different climate models, including a set of both surface and atmospheric (i.e. across several pressure layers) variables, such as wind speeds, temperature, and humidity. This foundational pre-training enables subsequent finetuning for various specific tasks.

**Architecture:** ClimaX extends the vision transformer architecture (Dosovitskiy et al., 2021) with novel encoding and aggregation blocks to accommodate the intricacies of weather and climate modelling and weather forecasting. Notably, it employs variable tokenization, treating each variable in the input as image-like data represented by a spatial map. These variables are then considered as separate tokens, enabling comprehensive analysis.

To tackle the challenge of scaling computations with an increasing number of input variables, ClimaX implements a variable aggregation technique. It performs a cross-attention operation for each spatial position across all variables, resulting in a unified token sequence that captures the holistic representation of the input data. The transformed data is then processed by a vision transformer in conjunction with a prediction head.

**Distinguishing Features from the Original Implementation:** In adapting ClimaX to ClimateBench, the authors introduced several modifications to their model differing from the weather forecasting stream. Firstly, only specific parts of the pre-trained weights from the foundation model were retained. This involved preserving and either freezing or finetuning the attention layers. Additionally, the ClimateBench implementation included a temporal aggregation model. This model processed each year of the 10-year input sequence through tokenization, aggregation, and attention layers, followed by global average pooling over the spatial map and cross-attention across time. The resulting embedded output was then fed into a linear attention head, mapping it to a spatial grid for a single variable and single time step only.

In contrast, our approach to climate projection diverges as we designed our task in a sequence-to-sequence manner. Thus, we maintain the same selection of pre-trained weights but exclude spatial map pooling and temporal aggregation, as our objective is to obtain predictions for each time-step in the input sequence. Additionally, we are modelling several output variables, i.e. temperature and precipitation, at the same time. Therefore, we utilize the original nonlinear decoder with a prediction head that maps the output to several variables across the spatial grid.

**Hyperparameters:** The hyperparameters for ClimaX were selected based on the original implementation and the author's adaptation to ClimateBench. The chosen values include a patch size of 16, embedding dimension of 1024, encoder depth of 8 with 16 attention heads, and a non-linear decoder comprising two MLP layers with GELU activation, followed by a linear prediction head layer. For training, we use an Adam optimizer with weight decay with a learning rate of 1e-3 and a weight decay of 1e-5 paired with a linear warm-up cosine annealing learning rate scheduler, with 5 warm-up epochs and a maximum of 50 epochs to keep it consistent with the training of other models. The warm-up start learning rate is set to 1e-8, with a minimum eta of 1e-8.

## K.2 U-Net

The U-Net architecture, originally proposed by Ronneberger et al. (2015), has established itself as a highly regarded convolutional neural network (CNN) model for image segmentation tasks. Its effectiveness in capturing intricate details while preserving the global context has made it a preferred choice for pixel-level classification and localization in various domains, including medical imaging and computer vision.

In a recent study conducted by Orlova et al. (2022), the potential of ML models for sub-seasonal forecasting of crucial climate variables was explored. The research encompassed an ensemble of models, incorporating linear models, regression models, random forests, and a U-Net-based convolutional network. The results demonstrated the superior performance of these ML methods over traditional non-ML baselines. Notably, the U-Net architecture exhibited exceptional capabilities in handling spatial variability within climate forecasts.

**Architecture:** The U-Net structure comprises an encoder path, responsible for capturing contextual information and extracting high-level features from the input image, and a decoder path, tasked with reconstructing the segmented image using the learned features. The decoder is followed by an output layer, typically employing additional convolutional operations to reduce the number of channels to match the desired number of output features.

**Distinguishing Features from the Original Implementation:** Following the implementation by Orlova et al. (2022), we adopt a U-Net architecture with a pre-trained backbone for the task of climate projection. Orlova et al. (2022) employed an off-the-shelf U-Net model available in the segmentation models PyTorch library. This pre-existing U-Net implementation utilized a pre-trained VGG11 (Simonyan and Zisserman, 2015) encoder backbone which was trained on the ImageNet classification challenge (Deng et al., 2009).

To adapt the U-Net model to our specific requirements, we encountered the challenge of dealing with input grids of varying resolutions. Unlike the original approach, we devised a solution by introducing zero-padding to the nearest image size divisible by 32 for both the longitude and latitude dimensions as this is required for the pre-trained encoder. Hence, we incorporated an adapted average pooling mechanism, initially introduced by Liu et al. (2018), to restore the output to its original grid size after the U-Net decoder process.

Furthermore, as our objective involved modelling sequence-to-sequence relationships rather than sequence-to-one, we employed a Time Distributed Layer encapsulating the pre-trained segmentation model. This approach, commonly used to handle sequential image data with convolutions, allows us to construct a sequential model with one layer per time-step in the input. Each layer consists of the same U-Net architecture, enabling the model to be applied to every temporal slice of the input. The weights are shared and trained simultaneously during the backward step.

**Hyperparameters:** Consistent with the original implementation (Orlova et al., 2022), the hyperparameters adopted include a VGG11 encoder, a linear readout layer, the Adam optimizer with a learning rate of 2e-4, and a weight decay of 1e-6. Additionally, an exponential learning rate scheduler with a gamma value of 0.98 is utilized for training.

## K.3 Convolutional LSTM

The Convolutional LSTM (CNN-LSTM) is a model that combines the sequential application of convolutional neural networks (CNNs) (LeCun et al., 1989) and long short-term memory (LSTM) (Hochreiter and Schmidhuber, 1997) networks. It is important to note that the CNN-LSTM should not be confused with an LSTM that employs convolutional operations for its gates. The motivation behind

the CNN-LSTM is to effectively process image-like data over time by extracting spatial dependencies using the CNN component and temporal dependencies using the LSTM component. Thus, the model can be used for climate projection tasks and has been recognized as the best performing baseline in terms of root mean squared error (RMSE) scores in ClimateBench (Watson-Parris et al., 2022), albeit with some spatial biases in its predictions.

**Architecture:** The architecture of the CNN-LSTM involves several key components. Firstly, the input data undergoes feature extraction through a CNN layer. To ensure the extraction of features at each time step while sharing the same module, a Time Distributed Layer is applied. Different weights are assigned to the filters in this layer. Following the feature extraction, an average pooling operation is performed to reduce the spatial dimensionality of the extracted features. This is also done time-step wise. Subsequently, the features are fed into a single LSTM layer, which captures the temporal dependencies in the data. Finally, a linear readout layer is applied to obtain the desired output.

**Distinguishing Features from the Original Implementation:** In comparison to the original implementation in ClimateBench, several modifications have been made to fit our training pipeline and our task setting. The ClimateBench implementation did not employ a sequence-to-sequence approach, only considering the last output of the LSTM and also did only model one output variable at a time. Contrary to that, in our modified version, all outputs of the LSTM are taken into account, allowing for modelling of the entire sequence. Thus, the readout layer has been altered to enable the modelling of the complete sequence and also to enable modelling multiple output variables simultaneously. Furthermore, the implementation has been converted from TensorFlow to PyTorch, which may result in slight changes due to the differences in internal workings of the layers between the two different libraries.

**Hyperparameters:** Regarding the chosen hyperparameters, we have adhered to the settings selected in ClimateBench (Watson-Parris et al., 2022). The CNN component consists of a single convolutional layer with 20 filters, each with a kernel size of 3, and employs ReLU activation function. The pooling stage involves two 2D average pooling layers, one with a kernel size of (2,2) and the other with a kernel size of (longitude/2, latitude/2). The LSTM layer comprises a single layer with 25 units and uses ReLU as the activation function. The model is trained using the Adam optimizer with a learning rate of 2e-4, a weight decay of 1e-6, and an epsilon value of 1e-8. Additionally, an Exponential Decay Learning Rate Scheduler with a gamma of 0.98 is applied during training.

### K.4 Gaussian Process

Gaussian process regression (Williams and Rasmussen, 2006) is a nonparametric and Bayesian regression method that has the advantage of providing uncertainty measures over predictions. Since we deal with large datasets, we use a stochastic variational variant of the Gaussian process for regression (SVGP; (Hensman et al., 2015)) that supports training with minibatches and scales well with the number of samples. To deal with the multi-output nature of the task, we use the Linear Model of Coregionalization (LMC) with 100 latent Gaussian processes. The number of latents used is a hyperparameter that controls the capacity of the model. We use Matern1.5 kernels and train with the Adam optimizer with a learning rate of $0.1$ and batch size of $64$.

As in ClimateBench (Watson-Parris et al., 2022), we use Empirical orthogonal functions (EOF) as a dimensionality reduction technique on the aerosols input (BC and SO2) and keep only the 5 first modes. While we use SVGP, the ClimateBench uses exact Gaussian Processes which can be problematic for large datasets since it scales cubicly with respect to the number of examples.

Finally, to implement our SVGP method, we used the library GPytorch (Gardner et al., 2018) (`https://gpytorch.ai/`), a Pytorch implementation of Gaussian processes.

We showcase our results with the Gaussian Process on six climate models in Table 4. Gaussian Process is the best-performing model when predicting TAS (surface air temperature) for NorESM2-LM. Generally, we observe that the Gaussian Process performs really well for predicting TAS but when predicting PR (precipitation), it underperforms the models shown in Table 5 by a huge margin. For this reason, we chose to run the Gaussian Process on a smaller subset of the climate models and omitted it for the experiments run on 15 climate models.

|  | GP | |
| --- | --- | --- |
|  | **TAS** | **PR** |
| **AWI-CM-1-1-MR** | 0.264 | *0.538* |
| **BCC-CSM2-MR** | 0.247 | *0.536* |
| **EC-Earth3** | 0.279 | *0.580* |
| **FGOALS-f3-L** | 0.280 | *0.553* |
| **MPI-ESM1-2-HR** | 0.257 | *0.544* |
| **NorESM2-LM** | **0.248** | *0.553* |

Table 4: RMSE of Gaussian Process single-emulating temperature (TAS) and precipitation (PR) on six climate models. Emboldened values are showing when the GP performs best across all ML models (GP, U-Net, ConvLstm, ClimaX, and ClimaX$_{\text{Frozen}}$). Italic values show when the GP performs worst across all ML models.

## L    Experimental Setup

### L.1    Single-Emulator

For all our models, we use a similar training procedure where we predict the outputs for entire input sequence in one go. Our inputs are of shape $< batch, sequence\_length, num\_vars, lon, lat >$ and outputs are of size $< batch, sequence\_length, 2, lon, lat >$ where the 2 in output dimension denotes TAS and PR. For our experiments we have a sequence length of 12, meaning that we use data from 12 months and predict the output for those 12 months. More details about the input and output data, along with the variables used are in Section 2.1. Our training methodology differs slightly from those used by Nguyen et al. (2023) and Watson-Parris et al. (2022) in the sense that we don't use a lead time for out models.

We train all our models on one Nvidia-RTX8000 GPU with a batch size of 4 and make use of 32GB of RAM. The U-Net and ConvLSTM are trained for 50 epochs with an initial learning rate of $2e-4$ with an exponential decay scheduler. To keep the training consistent for ClimaX models, we train the models for 50 epochs with an initial warm-up for 5 epochs. The learning rate for the warm-up is set to 1e-8 and then to 5e-4 for training. Other training details are kept the same as in the original implementation. Fig. 6 reports results from experiments without warm-up, while all tables in the Appendix report only results from experiments with warm-up. For the loss, we use the latitude-longitude weighted mean squared error (LLMSE) as implemented in Nguyen et al. (2023). We report the latitude-longitude weighted root mean squared error (RMSE) as our metric to evaluate the models.

### L.2    Super-Emulator

We use the term super-emulator to refer to a single ML model trained on the entire data of all climate models, such that it is capable of projecting a unique output of every specific climate model, according to a user requirement. Training a single ML model (as opposed to many ML models each trained on one climate model data) may benefit from rich and diverse data from the collection of climate models (as opposed to only one). This joint knowledge results in a super-emulator model with parameter sharing and increased capacity. For this goal, the super-emulator should be able to distinguish between data inputs provided to it from different climate models. A naive super-emulator that is trained on multiple climate models as targets without any contextual information would be mapping the same inputs to multiple targets. This lack of context restricts the emulator's ability to accurately represent the different behaviours of climate models. In order to prevent the "one-to-many" mapping, a super-emulator needs to encode the climate models in the input data. One approach is to provide which climate model is associated with each sample, and another approach is to provide the climate model's internal dynamics as additional input. Both methods are outlined in the following:

**Multi-head decoder:** This method is centred around the idea of indexing each sample with its respective climate model. By associating each sample with a climate model, the mapping becomes a

"one-to-one" mapping. Here, we implement this by using a multi-head decoder on top of a neural network model. We are using the exact same NN models as in the single-emulator experiments as basemodels. They use the same inputs, outputs, and shapes as in the single-emulator experiments. On top of such a basemodel (U-Net, ClimaX, ConvLSTM), a multi-head decoder is installed. Each climate model that participates in the super-emulator receives its own head. A head consists of several convolutional layers which can be determined by the user. Each head receives the gridded output of the basemodel *and* the climate model index for each sample. The training batches can contain samples from different climate models since each sample can be identified by its climate model index. Each sample is forwarded only through its matching head, and back-propagated through this head, while all other heads stay frozen. In our experiments, we used the ConvLSTM, U-Net, ClimaX, ClimaX$_{Frozen}$ as base models with the same training regimes as outlined in the single-emulator experiments. Due to memory requirements, we were only able to train the super-emulator on the smaller set of six climate models. [7] The overall super-emulator was trained for a maximum of 100 epochs with a head for each climate model and each head had 2 convolutional layers with 32 units. We noticed that the performance of ClimaX-based models plummeted and other models stagnated after $\sim$20 epochs with predictions converging to zero. This lead us to include a standard Early Stopping Regularization paradigm that stopped the training process around that threshold. We used the inbuilt pytorch Early Stopping class with patience set to 3 and a min delta of 0. Refer to Appendix M.2 for the results of those experiments. Further experiments are necessary to be able to analyze the potential of the multi-head decoder for super-emulation fully.

**Sea-level pressure maps:** The current approach of training a super-emulator solely on greenhouse gas, aerosols and aerosol precursor emission data from Input4MIPs does not sufficiently capture the intrinsic dynamics of the climate models. Climate models are not solely influenced by external factors but also by intrinsic dynamics, which can exhibit chaotic behaviour. Consequently, even numerical models generate different trajectories when provided with the same input data but different initializations. This internal variability is captured by the multiple ensemble members that are also included in ClimateSet. However, the current machine learning emulators fail to account for and explain this internal variability, resulting in fluctuations in the predicted targets. To address these challenges, one option is to incorporate one of the intrinsic dynamics present in climate models as an additional input alongside the forcing data. For instance, concatenating sea level pressure maps specific to the climate model and/or ensemble member with the input GHG and aerosol data would enable the required one-to-one mapping and partially explain the variability observed in the targets. This approach has the potential to enhance the emulator's overall performance. It would also enable querying for average predictions or interpolation between existing scenarios during inference time. By interpolating or averaging sea level pressure maps from existing runs, it becomes possible to generate new input features that incorporate both emissions data and intrinsic dynamics, even though the intrinsic dynamics have not been simulated by a climate model yet. Consequently, the machine learning emulator - despite using the output of a climate model - can still predict scenarios that the climate model itself has not run.

To summarize, the possibilities to design a super-emulator are plentiful and are subject to the intended purpose determined by the user. As such, they remain to be investigated thoroughly.

### L.3 Generalization Capabilities of ML Models

In order to investigate the generalization capabilities of ML models on different climate models, we decided to choose a finetuning approach.

### L.3.1 Finetuning Single-Emulators

For the single-emulator finetuning experiments, each ML model is first pre-trained on a specific climate model, and subsequently finetuned and evaluated on NorESM2-LM (Seland et al., 2020). The best model to be chosen for subsequent tuning is selected based on the validation metric. We chose NorESM2-LM as a generalization test case since it is the model that has been most widely used in previous work. We compare the performance of the finetuned ML model with the NorESM2-LM single-emulator to learn A) how well the ML model can generalize from one model to another, and B) how much transferable information each climate model has for NorESM2-LM. Similar to (Nguyen

---

[7]We plan to address this limitation in future work and will update our GitHub repo continuously.

et al., 2023), we hope that pre-training on climate datasets might help the ML model learn some patterns that are applicable to other datasets & climate models.

The single-emulator finetuning experiments follow the training procedure described in Section L.1. Those setups are used for pre-training. The pre-trained models are then loaded and afterwards finetuned on NorESM2-LM for 50 epochs. We report the latitude-longitude weighted root mean squared error (RMSE) as our evaluation metric.

### L.3.2 Finetuning Super-Emulators

The multi-head decoder used for super-emulation can also be used to examine the generalization capability of a super-emulator. For pre-training a multi-head decoder super-emulator is used with heads for all participating climate models. The only climate model that has no header is the one on which the super-emulator is evaluated. During finetuning another header is added and only that head is finetuned. Subsequently, the performance of the super-emulator (e.g. trained on a set of climate models but without NorESM2-LM) can be compared with the performance of the finetuned super-emulator (e.g. finetuned on an added head for NorESM2-LM). We leave those experiments and analyses for future work.

## M Experimental Results

### M.1 Single-Emulator

The results of our experiments for training ML models on emulating single climate models are shown in Table 5. In these experiments, we train our models individually on 15 different climate models and report the RMSE for our two output variables, TAS (surface air temperature) and PR (precipitation).

Our experiments indicate a trend of larger ML-models outperforming the smaller ones on single-emulation. For most climate datasets, ClimaX performs much better than other models. This is not unexpected as ClimaX is a much larger model being finetuned on suitable pre-trained weights (stemming from ERA5 and two climate models) for this task. ClimaX$_{Frozen}$ follows usually with slightly worse RMSE values. Thus, finetuning the weights is indeed helpful to increase the performance of ClimaX. With some exceptions, U-Net is performing best after ClimaX$_{Frozen}$, and ConvLSTM, the simplest and smallest model, shows often the worst performance across all ML-models. We conclude, that there is a general trend among the ML-models performance across several climate models.

However, there are exceptions to this general "leaderboard". We can identify three strong outliers, namely GFDL-ESM4 (Dunne et al., 2020), NorESM2-LM (Seland et al., 2020) and NorESM2-MM (Seland et al., 2020), where U-Net outperforms both ClimaX models by a pretty large margin. Also on TaiESM1 (Wang et al., 2021), U-Net is performing exceptionally well and comes in second after ClimaX. These results show, that it is not sufficient to evaluate an ML-emulator on a single climate model. When trained only on e.g. NorESM2 models, one might conclude that U-Net is the best performing model. Only when taking a set of climate model datasets intro consideration, we can evaluate how well a specific ML-model can emulate climate models.

Also note, that the performance differences between the ML models is not huge. For example, U-Net performs in a very similar range like ClimaX$_{frozen}$, outperforming it from time to time. This indicates that simple baselines can still keep up with larger models. We hope that other simple baselines will be investigated by other researchers in the future. Also further work on large models is needed to achieve performance gains that justify using such models over simple baselines.

### M.2 Super-Emulator

Our super-emulation experiments showed that the ranking of the best performing ML models changes between single and super-emulation. While U-Net and ClimaX have been the best performing models when the goal is to emulate only one climate model (see Table 5), ConvLSTM is performing best when a set of climate models needs to be emulated (see Table 6). The ConvLSTM achieves the highest performance among all ML models for all six climate models on both TAS and PR. However, all super-emulator performance values are significantly worse than the single-emulator results. This

|  | U-Net | | ConvLSTM | | ClimaX | | ClimaX$_{\text{Frozen}}$ | |
| --- | --- | --- | --- | --- | --- | --- | --- | --- |
|  | TAS | PR | TAS | PR | TAS | PR | TAS | PR |
| **AWI-CM-1-1-MR** | 0.321 | 0.326 | 0.445 | 0.438 | **0.263** | **0.267** | 0.316 | 0.321 |
| **BCC-CSM2-MR** | 0.232 | 0.258 | 0.295 | 0.333 | **0.205** | **0.229** | 0.232 | 0.258 |
| **CAS-ESM2-0** | 0.317 | 0.315 | 0.393 | 0.394 | **0.226** | **0.226** | 0.272 | 0.276 |
| **CNRM-CM6-1-HR** | 0.315 | 0.309 | 0.379 | 0.370 | **0.250** | **0.245** | 0.289 | 0.286 |
| **EC-Earth3** | 0.292 | 0.295 | 0.397 | 0.398 | **0.240** | **0.241** | 0.287 | 0.292 |
| **EC-Earth3-Veg-LR** | 0.299 | 0.300 | 0.410 | 0.410 | **0.248** | **0.250** | 0.290 | 0.290 |
| **FGOALS-f3-L** | 0.286 | 0.305 | 0.362 | 0.381 | **0.260** | **0.274** | 0.294 | 0.313 |
| **GFDL-ESM4** | **0.481** | **0.482** | 0.505 | 0.503 | 0.542 | 0.544 | 0.630 | 0.636 |
| **INM-CM4-8** | 0.232 | 0.235 | 0.335 | 0.346 | **0.212** | **0.214** | 0.257 | 0.261 |
| **INM-CM5-0** | 0.258 | 0.264 | 0.329 | 0.345 | **0.207** | **0.209** | 0.247 | 0.252 |
| **MPI-ESM1-2-HR** | 0.31 | 0.317 | 0.406 | 0.415 | **0.225** | **0.229** | 0.258 | 0.265 |
| **MRI-ESM2-0** | 0.292 | 0.292 | 0.357 | 0.360 | **0.221** | **0.219** | 0.262 | 0.262 |
| **NorESM2-LM** | **0.271** | **0.291** | 0.299 | 0.315 | 0.369 | 0.381 | 0.386 | 0.413 |
| **NorESM2-MM** | **0.291** | **0.313** | 0.317 | 0.333 | 0.376 | 0.395 | 0.420 | 0.435 |
| **TaiESM1** | 0.196 | 0.197 | 0.333 | 0.346 | **0.194** | **0.195** | 0.232 | 0.237 |

Table 5: RMSE values for each ML model single-emulating surface air temperature (TAS) and precipitation (PR) across 15 climate models. All reported values are averages over three seeds. The highest RMSE among the ML-models is emboldened for each climate model dataset. For most of the climate model datasets, ClimaX seems to outperform other ML models. U-Net is the best performing model on some of the climate model datasets.

|  | U-Net | | ConvLSTM | | ClimaX | | ClimaX$_{\text{Frozen}}$ | |
| --- | --- | --- | --- | --- | --- | --- | --- | --- |
|  | TAS | PR | TAS | PR | TAS | PR | TAS | PR |
| **AWI-CM-1-1-MR** | 0.579 | 0.571 | **0.396** | **0.398** | 0.908 | 1.172 | 0.647 | 0.634 |
| **BCC-CSM2-MR** | 0.334 | 0.299 | **0.270** | **0.238** | 0.495 | 0.88 | 0.366 | 0.338 |
| **EC-Earth3** | 0.508 | 0.481 | **0.287** | **0.286** | 0.616 | 1.287 | 0.491 | 0.480 |
| **FGOALS-f3-L** | 0.811 | 0.745 | **0.314** | **0.287** | 0.665 | 1.622 | 0.464 | 0.439 |
| **MPI-ESM1-2-HR** | 0.8 | 0.756 | **0.295** | **0.289** | 0.903 | 0.979 | 0.531 | 0.512 |
| **NorESM2-LM** | 0.566 | 0.532 | **0.261** | **0.249** | 0.686 | 1.457 | 0.455 | 0.426 |

Table 6: RMSE values for each ML model super-emulating surface air temperature (TAS) and precipitation (PR) on a set of six climate models. All reported values are averages over three seeds. The highest RMSE among the ML-models is emboldened for each climate model dataset. The performance of super-emulator models seems to be worse as compared to the single-emulator models as seen in Table 5. The smallest model, ConvLSTM, is performing best across all climate models. Smaller ML models seem to be the better super-emulators in our experiments.

can be attributed to the fact that the task of learning to emulate several climate models is harder than learning to emulate only one climate model.

Initially, we suspected that ConvLSTM outperforms the other models since the loss curves looked like it is able to learn "faster" due to its smaller amount of learnable parameters. However, this was only true when training on a small subset of the years. Further experiments covering the whole time-length showed that all ML models are early-stopping around 20 epochs as their performance gets worse after that. It should be noted, that ClimaX's performance is plummeting suddenly after 20 epochs. Further experiments could be done to try to stabilize the training and investigate if ClimaX performance improves over a longer period of training time. Future work could investigate different training regime settings for all models. The experiments we conducted to date indicate to us that all ML models need around the same amount of epochs to learn to super-emulate.

We think that the determining factor of the super-emulator performance is the models generalization capability. The better it can generalize onto different climate models, the better it can perform as a super-emulator. To investigate this, we conducted finetuning experiments (see M.3. They show that the smaller models are better at generalizing across different climate models, with ConvLSTM clearly outperforming all the other models. The ML-model ranking of the finetuning results aligns with the ranking for super-emulation. These results support our hypothesis that smaller models are better at generalizing and, for this reason, might be the better super-emulators for now.

We are convinced that adding challenging tasks - such as the super-emulation task presented here - will be helpful to benchmark ML models on climate relevant tasks.

## M.3 Generalization Capabilities of ML Models

The results for the finetuning experiments of the single-emulator are shown in Table 7. The first column lists the choice of the pre-training dataset where "None" means that the model is trained directly on NorESM2-LM (single-emulator experiment). ConvLSTM seems to outperform all other ML models for these experiments. One possible explanation for this is that ConvLSTM is indeed the ML model that can generalize best from one climate model to another climate model. Another reasonable explanation however is, that ConvLSTM is just *in general* the best ML model to emulate this specific climate model, NorESM2-LM. Hence, it is more informative to look at the performance differences between single-emulator and finetuned model which is shown in Table 8.

Here, we can see that the ConvLSTM achieves the largest performance gain through the finetuning setting, ClimaX$_{Frozen}$ gains slightly, U-Net neither gains nor looses, while ClimaX drops in performance. There are some exceptions, e.g. ClimaX gets the highest performance gain for FGOALS-f3-L. The slight performance gain of ClimaX$_{Frozen}$ could be explained by the fact that it had the poorest overall performance on NorESM2-LM before, i.e. there was more space for improvement compared to e.g. the U-Net that was already performing very well on NorESM2-LM. However, the pretty consistent performance gain of the ConvLSTM cannot be explained by that, indicating that the ConvLSTM could actually be inherently better at generalizing to other climate models. While ConvLSTM seems to benefit from pre-training, it makes less of a difference for U-Net, and seems to have little benefit for ClimaX when evaluating on the NorESM2-LM dataset. The benefit for ClimaX might be limited as compared to other models as it has already been pre-trained on a collection of datasets as described in Nguyen et al. (2023).

Overall, Table 8 shows that finetuning on one climate model can improve the performance on another model. This indicates, that climate models indeed share patterns and information and that patterns learned from one climate model can be transferred to another.

| Pre-training Dataset | U-Net | | ConvLSTM | | ClimaX | | ClimaX$_{Frozen}$ | |
|---|---|---|---|---|---|---|---|---|
| | TAS | PR | TAS | PR | TAS | PR | TAS | PR |
| None | 0.291 | 0.309 | **0.278** | **0.292** | 0.311 | 0.325 | 0.353 | 0.370 |
| AWI-CM-1-1-MR | 0.302 | 0.325 | **0.208** | **0.220** | 0.351 | 0.365 | 0.335 | 0.346 |
| BCC-CSM2-MR | 0.327 | 0.253 | **0.277** | **0.291** | 0.354 | 0.368 | 0.326 | 0.344 |
| EC-Earth3 | 0.302 | 0.319 | **0.247** | **0.262** | 0.385 | 0.399 | 0.329 | 0.343 |
| FGOALS-f3-L | 0.283 | 0.306 | **0.242** | **0.256** | 0.327 | 0.341 | 0.320 | 0.342 |
| MPI-ESM1-2-HR | 0.266 | 0.284 | **0.218** | **0.229** | 0.412 | 0.427 | 0.304 | 0.317 |

Table 7: RMSE values for ML models that were finetuned on **NorESM2-LM** and pre-trained on different climate models. We report the RMSE for two variables, TAS (surface air temperature) and PR (precipitation). The best performing models are emboldened. The pre-training dataset column shows which dataset the model was initially trained on before being finetuned on the NorESM2-LM dataset. The first row with "None" for the pre-training dataset represents the results from training on NorESM2-LM from scratch. ConvLSTM performs best across all climate model datasets here.

| Pre-training Dataset | U-Net | | ConvLSTM | | ClimaX | | ClimaX$_{Frozen}$ | |
|---|---|---|---|---|---|---|---|---|
| | TAS | PR | TAS | PR | TAS | PR | TAS | PR |
| AWI-CM-1-1-MR | -0.011 | -0.016 | **0.070** | **0.072** | -0.004 | 0.004 | 0.089 | 0.024 |
| BCC-CSM2-MR | -0.036 | **0.056** | **0.001** | 0.001 | -0.043 | -0.043 | 0.027 | 0.027 |
| EC-Earth3 | -0.011 | -0.016 | **0.031** | **0.030** | -0.074 | -0.074 | 0.024 | 0.027 |
| FGOALS-f3-L | 0.019 | 0.013 | 0.005 | 0.006 | **0.058** | **0.058** | 0.009 | 0.001 |
| MPI-ESM1-2-HR | 0.025 | 0.025 | **0.060** | **0.063** | -0.101 | -0.102 | 0.049 | 0.053 |

Table 8: RMSE delta values for ML models that were finetuned on **NorESM2-LM** and pre-trained on different climate models. Here, we report the RMSE differences between pre-training and finetuning for TAS and PR. The pre-training dataset column shows which dataset the model was initially trained on before being finetuned on the NorESM2-LM dataset. Positive digits indicate that pre-training on this climate model improved performance compared to the single-emulator experiment, negative digits that performance dropped. ConvLSTM is the model that can generate the largest performance gain by finetuning.

