# OpenReview forum: "ClimateSet: A Large-Scale Climate Model Dataset for Machine Learning"
_NeurIPS.cc/2023/Track/Datasets_and_Benchmarks — NeurIPS 2023 Datasets and Benchmarks Poster_

### Official Review · Reviewer_fdkb · 2023-07-06
**Official comments**

**Rating:** 7
**Confidence:** 2
**Correctness:** Yes. Yes. Yes.

**Strengths:**

The dataset, especially its benefit, is unique, details listed in Summary And Contributions.

**Additional Feedback:**

Please see Summary And Contributions.

**Clarity:**

The paper is easy to follow (for the ML part, as the climate part took up some of my time).

**Documentation:**

Yes.

**Ethics:**

None.

**Limitations:**

The authors have discussed a lot in the paper.

**Opportunities For Improvement:**

More ML-related analysis may make the paper more attractive, e.g., the potential spurious correlation mentioned in Summary And Contributions.

**Relation To Prior Work:**

The authors have discussed a lot in the paper.

**Summary And Contributions:**

This work aims to introduce a large-scale climate dataset to the ML community built upon the proposed pipeline, consistent with AI's aim for science. The authors show the benefit of the constructed dataset, e.g. much bigger and practical than previous ones.

This work is promising. However, I have a concern about the constructed dataset. Specifically, the context keeps changing due to the interaction between climate and various factors, thus the selected data may omit key factors that have direct or indirect impacts on the climate. Namely, even the same observation climate data may have a different "future" due to these factors. This is a multiple-output problem, i.e., a one-to-many problem. A simple strategy is to study the spurious correlation between the ground truth and the input, as it is shown that these two variables typically relate to each other [1].

I have to claim that I have limited background knowledge about the climate field. But this work is exciting and gives me the attempt to have a try. Thus, as a no-person in the community, I give a rating of 6 due to the promising future of this field.

[1] CausalAdv: Adversarial Robustness Through the Lens of Causality, Zhang et al. ICLR 2022

---

> ### Author Response · Authors · 2023-08-23
> **Response to review**
>
> Thank you for taking the time to read and review the paper so thoughtfully, especially given the limited background you note with climate science. We appreciate your feedback and find your point about the one-to-many problem very interesting, as we have discussed this also among ourselves.
>
> **The one-to-many problem:**
> In the single-emulator case, it is not really a problem, since each climate model is a deterministic map that takes an emission scenario as input and outputs a climate future - that is, the input to the model already captures uncertain factors such as how much CO2 the world emits. Different inputs to climate models do of course lead to different outputs, as you note. When using multiple climate models this would be a potential problem, since the same input (one GHG scenario) leads to different outputs that are all possible (different climate model outputs). There are two possible ways to mitigate this:
>
> (1) Providing information of which climate model or ensemble member is used, as we do in our superemulation experiments. A promising direction for future work is to learn more complex encodings of the different climate models or ensemble members that can implicitly capture the “distance” between the different predictions they make).
>
> (2) Using sea-level pressure maps as an additional input. This would serve as an independent variable (not affected by climate change). The sea level pressure map would represent/encode the specific workings/processes of the underlying climate model and we could map from emissions + sea level map to the climate model’s output. The sea level pressure map would look different for each model and ensemble member.
>
> We have extended our discussion of super-emulation in the appendix (Appendix K3) to discuss this question in more detail.
>
> Please let us know if there are further questions that we can help to address.

---

> > ### Author Response · Authors · 2023-08-30
> > **Response to Reviewer fdkb**
> >
> > Dear reviewer,
> >
> > Thank you again for taking the time to review our work. Let us know if you have any concerns left after our response. We would be happy to discuss any further questions and comments you may have. Please let us know if you have received our responses and if we have successfully addressed your concerns. Thank you once again for your feedback.

---

### Official Review · Reviewer_NfhR · 2023-07-21
**Review of ClimateSet, a suite of climate model simulation data for training ML emulators**

**Rating:** 6
**Confidence:** 3
**Correctness:** Claims seem correct in my opinion. Re…

**Strengths:**

The main strength of the paper is in its expansion on the number of climate models used in the dataset creation with a careful consideration on aspects such as variable availability, spatiotemporal resolution normalization, and aligning inputs (forcing/boundary conditions) to the climate models with the output predictions as well as data cleaning. The number (and diversity) of the climate models used could be very important towards building foundation models that can emulate different climate scenarios and is highly relevant to machine learning researchers in the climate science domain. Unlike weather forecasting datasets, the most famous being ERA5 re-analysis dataset, climate model datasets could be far larger owing to different scenarios, and there is a lack of a comprehensive dataset that spans multiple models and warming scenarios in this space today.

**Additional Feedback:**

-Some condensed information could be provided on the machine learning task in the main paper. Climate models are free-running simulations from an IC with forced boundary conditions. Is the machine learning emulator similar? What is the exact task (as I understand it, its predicting temperature+precipitation from boundary conditions), is historical information included, is it a diagnostic-type prediction (same time step) or an autoregressive prediction?
-Are all climate models initialized (for the state variables T, precip) the same way? And is this information also provided to the ML models.
-The super emulator is talked about as a major contribution but do not see results (unless I missed it somewhere).

**Clarity:**

Paper is reasonably clear. Some content/information is repetitive in the beginning sections (into, datasets) and could be more streamlined/shortened. Space could be used to show more plots on the difference in the datasets and maybe also show the emulation of some critical events in the climate model simulations through the ML models.

**Documentation:**

Yes.

**Ethics:**

None.

**Limitations:**

Authors talk briefly about model diversity and similarities amongst models as a limitation in the appendix. Also see improvement opportunities above.

**Opportunities For Improvement:**

1. While the number of climate models is large, the resolution still seems quite coarse at 100/250km. The authors talk about things like "extreme event prediction" but is that resolution sufficient? Storm resolving processes are at km scale resolution and uncertainties in climate models occur through subgrid parameterizations (<km). The temporal resolution (1 month) also seems coarse to predict things like heat waves in the future or frequency of cyclones etc. Can the dataset be enhanced with higher resolution models?
2. What about other variables? temperature and precipitation are quite important but what about other prognostic variables? I assume, like weather forecasting, climate models forecasts 100s of variables (at different pressure levels).
3. It is unclear if the models chosen represent a sufficient diversity amongst different scenarios. I do not know much about the different greenhouse gas emission scenarios and while Fig 1 does show qualitative changes, it would be nice to see other metrics that demonstrate quantitatively what has changed in the predictions. For instance, pdfs of surface temperature and precipitation across different scenarios or spatial averages across several years.

**Relation To Prior Work:**

Yes.

**Summary And Contributions:**

This paper presents ClimateSet, a dataset with model simulations spanning a large historical record as well as future climate simulations from different forcing/boundary conditions that forecast the future earth's climate under different climate change scenarios. The intention of the datasets is to train either climate change scenario-specific ML models or a "super emulator" that can learn the different scenarios. The paper explains the dataset generation process, lists the models used and the difficulty in curating fields from different models and scenarios and also trains some baseline ML models (CNNs and transformers) and shows their capabilities on the different scenarios.

---

> ### Author Response · Authors · 2023-08-23
> **Response to ClimateSet Review**
>
> Thank you very much for taking the time to read and review our work in detail. We found the questions you brought up particularly helpful to improve our manuscript and we adapted it accordingly. Below you can find our responses to your comments:
> - **(1a) Climate model resolution**: Yes, the ClimateSet pipeline does indeed provide the functionality to retrieve climate models with higher resolution. ClimateSet was built exactly for this purpose. The user is only limited by the limits of the climate model itself and the availability of the model and the desired parameters on ESGF. To date, only very few climate models go beyond the resolution of 100km/250km that we provide in our core dataset. Due to the high computational demands, climate models with higher resolution than 100 km (part of HighResMIP) have to trade-off at other points, e.g. they cannot provide data for all 4 main SSP scenarios and have fewer ensemble members.
> This is why downscaling climate model data is such a high-impact field and we hope that ClimateSet will also be used to address downscaling climate models. To summarize, the user can both request higher temporal resolution (available), and higher spatial resolution if available. We adapted the relevant section in the manuscript to make this more explicit.
> - **(1b) Extreme events**: . We are looking at extreme events not on a local, but rather on a global scale. On the temporal dimension, the events are extreme in comparison to previous decades (e.g. not the hottest day of the year, but rather the hottest decade of the century). On the spatial scale, we do not look at single cyclones, but rather e.g. heat waves of a climatic region. The extreme events are thus relative to the resolution we have. This information is still crucial, also for extreme event prediction on a much finer scale: To predict future extra-tropical cyclone behaviour we need to know the temperature gradients of our future (where, when, and how often), especially those that are extreme in relation to the temperature gradients we face nowadays. Thank you very much for calling our attention to the lack of clarity in the text on this point - we decided to remove it as an example since it would need further clarification as we provided it here.
> - **(2) Other variables**: Yes, absolutely, ClimateSet can be extended with any variable available of a climate model through the pipeline. And yes, climate models usually cover hundreds of variables.
> - **(3) Diversity of SSP scenarios**: Thank you for calling our attention to this. We included a Fig. showing the range of temperature projections (across SSP scenarios, min, mean, max) of 4 climate models for each year. This shows the differences between the climate model quantitatively and not only qualitatively. We also guide the reader to a source that more thoroughly explains SSP scenarios (footnote introduction, IPCC glossary) and explains their differences in the Input4MIPs section.
> - **Clarity**: We condensed the information in the CMIP6 and Input4MIPs section (merging “Selection” and “Specifications”) and restructured the introduction to streamline it more. We also adapted the manuscript to avoid explaining climate model terms repetitively at different spots. We use the new space that resulted from condensing the information e.g. for an overview figure illustrating the emulation task.
> Machine learning task: We have added a description of the ML task in the section “Benchmarking Setup / Task”. The goal is to predict a time series of climatic variables (e.g. temperature and precipitation from 2015-2100) from a given parallel time series of forcing agents (GHG and aerosol emission maps from 2015-2100). We treat the task as a diagnostic-type prediction, however, it can also be treated as an autoregressive task. ClimateBench e.g. mixes the task settings depending on the model, while we prefer to keep the task consistent. Please let us know if you would add any further information.
> - **Climate model initialization**: Yes, climate models have different initializations. There are some commonalities encoded in the names of the different ensemble members. This means that two different climate models with the same ensemble member ID have a very similar initialization. This information is currently not explicitly given to the ML mode. It could certainly be interesting to provide the ML models with information about shared submodels of the climate models (e.g. two different climate models, but they have the same ocean model) as an auxiliary input, and we have added this to our discussion in future work (Appendix E). Thank you for the suggestion.
> - **Super emulator**: We reran the experiments since we updated our architecture and method for the super emulation. The revised results and the discussion of those results can now be found in the Appendix.
>
> Thank you for helping us improve the manuscript and our work. Please let us know if there are further questions that we can help to address.

---

> > ### Comment · Reviewer_NfhR · 2023-08-29
> >
> > Thanks to the authors for addressing all my concerns. I raise my score to 6.

---

> > > ### Author Response · Authors · 2023-08-29
> > > **Thank you**
> > >
> > > Dear reviewer,
> > > Thank you very much for your time and your comments and thus helping us to improve our work. We are glad that our revisions addressed all your concerns and thank you for raising your score!

---

### Official Review · Reviewer_jcks · 2023-07-21
**The work is meaningful for climate forecasting, but there is still room for improvement.**

**Rating:** 6
**Confidence:** 3

**Strengths:**

1 The dataset collects the output of different climate models from CMIP6, which can better reflect the differences of different models;

2 The ClimateSet data pipeline provided to access the dataset can preprocess the raw data without significant domain knowledge required;

3 The ClimateSet data pipeline provides a way to extend the core ClimateSet dataset.


**Additional Feedback:**

The title of the article--ClimateSet: A Large-Scale Climate Model Dataset for Machine Learning may be misunderstood. The data set proposed in the article is more focused on the simulation of the outputs of climate model in CMIP6, rather than using the outputs of the climate model to make climate forecasts.

**Clarity:**

The paper is well written. It will be better if the article can further explain the technical terms about climate, so that readers in the ML community can better understand this work and realize the value of this work. For example, the paper can explain the definition of SSP126, SSP245, SSP 370, SSP585 and the differences among them.

**Correctness:**

The ClimateSet dataset is constructed in a sound way，and tools are provided to extend the dataset.

**Documentation:**

There is sufficient detail on data collection and organization, availability and maintenance, but no detail about ethical and responsible use.
The submissions include documentation and intended uses; the URL for accessing to the data; and a hosting, licensing, and maintenance plan.
The detail is not sufficient enough to support reproducibility, for example, how to use U-net model to forecast climate.


**Ethics:**

I don’t suspect there are any ethical concerns with the submission that warrant further discussion or review.

**Limitations:**

1	The submission doesn’t explain the rationality of using the climate model forecast data in CMIP6.
2	The submission doesn’t explain the rationality of choosing these 4 variables (CO2, CH4, SO2, BC) only in the core ClimateSet dataset.


**Opportunities For Improvement:**

1 The submission needs to explain the rationality of using the climate model forecast data in CMIP6 and why we cannot use observational or re-analysis data to address climate change related questions (line53-56). The output of climate model is also simulated from observational data and re-analysis data. There are also errors in the results of climate model simulations.

2 The function of CDO is not powerful enough, NCL can be added to ClimateSet preprocessor.

3 There could be more information about how to do benchmarking，so that the reader can reproduce the experiments in the article more easily.

4 The title of the article--ClimateSet: A Large-Scale Climate Model Dataset for Machine Learning is misunderstood. The data set proposed in the article is more focused on the simulation of the outputs of climate model in CMIP6, rather than using the outputs of the climate model to make climate forecasts.

5. The submission can explain why only the four variables mentioned were chosen in the core ClimateSet dataset，and BC is an aerosol not a GHG.


**Relation To Prior Work:**

This paper discussed clearly how it differs from pervious contributions，it adds the output of climate models besides observational and re-analysis data, supplies tool to preprocess data in different resolution and supplies a core dataset of ClimateSet consisting of 36 climate models rather than only one climate model.

**Summary And Contributions:**

The submission introduces ClimateSet dataset, which contains the inputs and outputs of 36 climate models from the CMIP6 and Input4MIPs archives. It also provides a modular dataset pipeline for retrieving and pre-processing additional climate models and scenarios. The submission also showcases the potential of ClimateSet by using it as a benchmark for ML-based climate emulation.

The contributions include:
1 This submission introduces the ClimateSet data pipeline, which can be used to retrieve and preprocess climate model data from CMIP6 and Input4MIPs for climate-related ML tasks;

2 This submission builds a core ClimateSet dataset with outputs of 36 climate models and inputs greenhouse gas emission fields for 4 different SSP scenarios and historical data;

3 This submission uses ClimateSet to compare state-of-the-art ML methods across data from 36 different climate models on a climate emulation task, and the results are both qualitatively different and more reliable than was possible in previous work.

---

> ### Author Response · Authors · 2023-08-23
> **Response to points of improvement**
>
> Thank you for reviewing our work and pointing us to where we can improve it. We attempt to address your specific comments below:
> - **(1) Climate model data vs reanalysis**: Thank you for pointing out that this is not communicated clearly enough in the paper. The rationale behind using climate model data and not observational or reanalysis data is that in order to simulate the climate we need to be able to make predictions under conditions different from those that have already occurred in the past. Reanalysis data is therefore limited, not merely in quantity, but also in the set of scenarios that are considered. Machine learning is famously bad at extrapolation, and therefore it is essential to ensure that, e.g., models are trained on data including higher GHG concentrations than occur at present. You are of course right that physics-based climate models have errors and are not perfect when compared with observational data, but they are the best tool that we have for working with future climate conditions, and therefore all prior work in ML for climate model emulation has used climate model data, rather than observational or reanalysis data. We have improved the discussion in the paper to reflect this important point.
> - **(2) CDO and NCL**: We have included a remark about NCL in the paper and cite it (Appendix H) and will make sure that our code is compatible with NCL extensions. We believe that at present CDO is preferable for all tasks we perform (even though NCL is more expressive). Namely, for tasks such as remapping, the NCL manual refers users to CDO since it is faster in this case.
> - **(3) How to do the benchmarking**: We agree that the benchmarking could be explained better and more user-friendly. We are working on Jupyter notebooks (GitHub), showing and explaining user-oriented how to perform benchmarking on ClimateSet.
> - **(4) Title**: We are sorry if this came across the wrong way. The paper is focused on providing a general dataset of climate models and we hope that the dataset will be used for a range of machine learning tasks that concern climate models (such as e.g. downscaling). Climate model emulation - the task presented here - is really just one task for which the dataset can be used. This is why we did not include the specific task in our title. The reason we did not include “CMIP6” in the title is because climate models used in ML and policy-making are almost always from the CMIP archives (since they are the “established/evaluated” climate models) and it makes the title more readable for readers unfamiliar with the term CMIP.
> - **(5) The 4 GHG / aerosols**: Agreed, we added those explanations.
> - **BC is not a GHG**: You are of course correct, we have corrected this.
> - **Climate terms (e.g. SSP definitions)**: For the ML community, a more detailed description of those terms would certainly be helpful. The terms CMIP6, SSP, and the different SSP scenarios are now explained in the introduction and Input4MIPs section. Furthermore, we added a footnote in the very beginning that links the reader to a glossary by the IPCC that we recommend for more in-depth explanations. Please let us know if there are other terms that you would have found useful to be explained.
> - **Ethical and responsible use**: We are unclear about how to act on your point that we have not documented the ethical and responsible use of our work. Could you clarify further? In our datasheet, we aimed to describe scenarios in which our work could be used in an unethical way (and state that this is not the intended use). If you do not find this section satisfactory, we would be happy to extend it if given more detail.
> - **Reproducibility (e.g. U-Net)**: We are writing for two audiences. One is an ML audience for which the intended use consists of performing as well as possible in terms of emulating the physics-based climate models. We hope that reproducibility for this audience is provided by our paper + source code, but would very much appreciate any feedback you have on how this can be improved. We are also working on Jupyter notebooks for this audience that make the benchmarking more accessible. The second audience is a climate science audience, who are the people who would be forecasting a future climate. Providing a guide to how to use climate models in such real-world scenarios is beyond the scope of a NeurIPS publication, as it is more domain-centric. Our goal is that our ML models can be used in place of standard climate models according to the standard practices followed by climate scientists, and it is with this goal in mind that we have matched the data to the full suite of climate models used in practice by climate scientists.
>
> Thank you once again for your time and thoughtful review. Please let us know if there are further questions that we can help to address.

---

> > ### Author Response · Authors · 2023-08-30
> > **Response to Reviewer jcks**
> >
> > Dear reviewer,
> >
> > Thank you again for taking the time to review our work. Let us know if you have any concerns left after our response. We would be happy to discuss any further questions and comments you may have. Please let us know if you have received our responses and if we have successfully addressed your concerns. Thank you once again for your feedback.

---

> > > ### Comment · Reviewer_jcks · 2023-08-30
> > >
> > > Thanks a lot for your efforts to response my question. Most of my concerns have been addressed.

---

> > > > ### Author Response · Authors · 2023-08-30
> > > > **Response to Reviewer jcks**
> > > >
> > > > Dear reviewer,
> > > > Thank you very much for your response. We are glad that we could address most of your concerns. Given the revised state of our work, would you consider raising your score?

---

### Official Review · Reviewer_MSxX · 2023-07-24
**solid paper that needs to include the dataset link and revisit their evaluation metric**

**Rating:** 6
**Confidence:** 3

**Strengths:**

- The dataset fills an important missing gap relative to prior dataset for this task -- it includes results from a large set of canonical climate models and collates them in a form amenable to ML prediction.
- The paper+appendix is quite detailed, which should make it easy for users to understand and interface with.
- The results illustrate that much progress remains to be made in the area, given that the new model (not the focus of this work) outperformed prior work. This is a strength because it should draw attention from other researchers who are looking to make an impact on this area.
- The data processing pipeline seems fairly general and potentially useful to other tasks.


**Additional Feedback:**

N/a

**Clarity:**

- The introduction should mention the significance of CMIP6 before L70 in order to clarify the significance of this claim.
- The acronym "SSP" should be spelled-out before L74.
- L76 It is not clear what super-emulation means.
- L203 Clarify the subjective statement "takes too long to be carried on a local machine that only supports single-threading for CDO" with an observed or estimated runtime.

**Correctness:**

- Claims are mostly correct, although the dataset link was redacted, so I could not verify the pipeline and dataset.
- Dataset is soundly constructed
- The benchmark is somewhat flawed. See my comment about RMSE evaluation.

**Documentation:**

No - the data link was redacted, so I could not access the data.

**Limitations:**

- Mostly, although one significant limitation is in the evaluation (see my comment about RMSE evaluation), yet this is not mentioned in the limitations. This is a limitation for both past work that relies on this metric and this paper's re-use of this metric, as the evaluation metric used in this paper is likely to become standardized for this dataset, yet it has important flaws.

**Opportunities For Improvement:**

- The data link was redacted, so I could not access the data. This makes it difficult to validate many of the claims.
- A table that lists the average performance of all models is needed (from which the result on L284 can be observed).
- L112 "2100" needs units (2100 A.D. I assume).
- S2.2. would benefit from a system diagram (e.g. a flowchart), consider moving it from the appendix into the main text.
- L242 The data shape tuple description would be improved with an example concrete shape to give readers a sense of the dimensionality of the modeling problem. Perhaps add a parenthetical statement with an example shape for a particular task.
- The evaluation metric in all tasks is RMSE. However, when one of the underlying simulations is stochastic, or in the 'super-emulation' task, in which a mixture of underlying simulators is used, the possible futures generated by these simulators may be multimodal, and therefore measuring a model's RMSE is not ideal, as it rewards models that generate answers at the centroids of the modes, rather than models that match the probability density function of the modes. The most general fix to this problem is to treat the problem accurately as a problem of conditional probability density estimation $p(\text{climate future} | \text{greenhouse gas and aerosol emission future})$$, require models to form conditional probability density estimates of events, and measure their log-likelihood of the event outcomes. This is especially important because of the inherent uncertainty of the problem, thus it is much preferred to treat the problem as one of assigning density estimates. The evaluation could be improved by adding results that correspond to converting every deterministic model (e.g. U-Net, ConvLSTM, etc.) to a stochastic one by creating a multivariate normal distribution centered at the current predictions, and assign all models the same covariance (e.g. an Identity matrix), and evaluate these models' average log-probability of the event outcomes. By doing so, it enables future ML models to be evaluated with the same metric, and such models might be made significantly better by being multimodal (e.g. a mixture of 2+ Gaussians).

**Relation To Prior Work:**

Yes

**Summary And Contributions:**

This paper proposes a dataset and dataset processing pipeline for climate emulation and related tasks. The dataset consists of the results of 36 existing climate models, which can be treated as training data for ML prediction tasks. RMSE evaluation results are presented on the tasks of temperature and precipitation responses, and a new simple UNet model was added and found to outperform prior work.

---

> ### Author Response · Authors · 2023-08-23
> **Data link and evaluation metrics**
>
> We appreciate your careful and deliberate feedback, especially regarding the evaluation metrics. In the following, we respond to all your comments.
> - **Data link**: We are sorry that you were not able to retrieve the dataset and apologize for not making the instructions more explicit. To keep the authors anonymous we did not provide the GitHub link that we will provide at the time of publication. For reviewers to have access to the data, we instead provided a script to download the core dataset. You can find instructions for downloading the dataset in “INSTRUCTIONS.txt” or “causalpaca_emulator/README.md” in the supplementary material.
> - **Table with average performance**: We included a list with the average performance of the ML models on 15 climate models in Appendix L.1. We refer to it shortly before L284. On GitHub we will provide the same list and extend it over time with more ML models and climate models.
> - **Units in L112**: Yes, absolutely, thank you.
> - **S2.2 system diagram**: We agree that S2.2 would benefit from a system diagram as provided in our appendix. Similarly, S3 could benefit from a figure illustrating the task. We included the overview figure for the latter for the broader audience and decided to include a simplified version of the current system flowcharts on our GitHub page for our users, so they have direct access to this visual information. If you have suggestions on how we could improve our current system diagrams/flowcharts further, we would be glad to hear them.
> - **L242 data shape tuple**: Great idea, thank you very much.
> - **Probabilistic metric**: Thank you very much for bringing this up. While we follow prior literature in treating this problem from a deterministic perspective, we agree that it could also be treated as conditional probability estimation. However, this is a subtle problem since using a stochastic ML model could entangle three different forms of uncertainty: 1) variability between the output of different climate models, 2) variability between different ensemble members within a single climate model, 3) uncertainty of prediction by the ML model itself. (Individual ensemble members within a climate model are deterministic, though this is an ongoing topic in the climate model community.)
> It is important if using stochastic methods that all three factors are handled transparently. A method that is mainly capturing (3) might lead the reader/user to think that it is capturing uncertainty about the climate future when it actually mostly captures the uncertainty of the ML model. While there is a strong need for probabilistic estimates, there is still considerable work that needs to be done before being able to do that. To address (1) properly, a weighting of the different climate models is necessary (since they share submodels, some are more advanced than others, etc.), and this is ultimately an open question for climate scientists to answer, not for the ML community. To address (2) properly we need to train the models on more than one ensemble member and for many models there are not enough ensemble members to capture the internal variability. Most approaches would mainly capture (3), which is not the intended outcome.
> Matching the centroids is therefore, in our opinion, sufficient for the moment and in line with prior work such as ClimateBench. But we agree it would be good to work towards stochastic models. We like your proposed approach to get probability estimates capturing (1) and (3) and would like to come back to it in our future work where we could solely focus on that topic. We will update ClimateSet’s GitHub repo accordingly.
> - **Limitation of evaluation metrics**: We agree this is a current limitation of the dataset, especially regarding a lacking probabilistic metric. We run our experiments on other metrics related to the longitude-latitude weighted RMSE we used (different weightings, simple RMSE and MSE). We did not add those metrics since they are closely related to each other. Similarly like ClimateBench we will provide an overview of the ML models' average performances on our GitHub page and we will continuously update those metrics and extend them. We also want to highlight that different climate variables might need different evaluation metrics. We would like to encourage the broader community to develop a set of evaluation metrics on that task that is informative for both the ML and the climate community. We added this discussion to our limitations and we are making sure that a follow-up on the set of evaluation metrics is possible.
> - **L70, L74, L76 (clarity)**: Accepted, thank you.
> - **L203 (CDO runtime, single-threading CDO)**: We mentioned in Section 2.3 (Usage - Accelerate ClimateSet) that the runtime of CDO in single-threaded mode is ~160h for 36 climate models. In L203 we are referring to this as a limitation for local machines. We have added a reference to make this link clearer.
>
> Thank you once again for your very helpful comments.

---

> > ### Author Response · Authors · 2023-08-30
> > **Response to Reviewer MSxX**
> >
> > Dear reviewer,
> >
> > Thank you again for taking the time to review our work. Let us know if you have any concerns left after our response. We would be happy to discuss any further questions and comments you may have. Please let us know if you have received our responses and if we have successfully addressed your concerns. Thank you once again for your feedback.

---

### Official Review · Reviewer_PvGq · 2023-07-27
**ClimateBench v2.0**

**Rating:** 8
**Confidence:** 3
**Correctness:** yes

**Strengths:**

This contribution fills a clear need and is obviously relevant to the broad climate change research community. The dataset also seems straightforward to use for climate science domain experts.

**Additional Feedback:**

n/a

**Clarity:**

For the most part. However, I do feel that something is missing that could make the dataset more user-friendly (particularly for non-experts and those not already familiar with ClimateBench). For example, while you mention in multiple places that GHG concentrations are inputs and the global temperature/precipitation are the outputs, I don't feel that it is obvious in the way that a figure or table would convey. Additionally, the description of the Input4MIPs section is a bit confusing. The ML task is that of climate model emulation, but the way the section is written makes it sound like the Input4MIPs data is not what was used in the actual simulation of the climate model data, which may confuse readers/users.

**Documentation:**

yes

**Limitations:**

Yes I am satisfied with the limitations section of the paper.

**Opportunities For Improvement:**

I don't have any large concerns or substantive suggestions for the product itself.

**Relation To Prior Work:**

Yes.

**Summary And Contributions:**

ClimateSet addresses a clear need from the community to have an easily accessible dataset for machine learning emulation of climate model predictions of global temperature and precipitation that uses a large suite of climate models rather than just one (as in ClimateBench). The dataset is indeed quite similar to the successful ClimateBench dataset but uses 36 models.

---

> ### Author Response · Authors · 2023-08-23
> **Response to ClimateBench v2.0**
>
> Thank you very much for your time and reviewing our work, we very much appreciate your feedback.
>
> In response to your comments on clarity:
> - **Input-output figure**: This is a good point. We added a more general figure that makes it easier to grasp the input and output immediately, providing an overview and sketching the emulator task.
> - **Input4MIPs section**: We have rewritten this section for clarity and hope that we convey better that the Input4MIPs data is indeed both our input and also the input to the climate models. The overview figure mentioned above will also help address this problem visually.
> - **User-friendliness**: We implemented your suggestions (see above) to make the dataset more user-friendly. We are also working on Jupyter notebooks (provided on GitHub) that will provide quick and comprehensible access to benchmarking ML models on ClimateSet.
>
> Thank you again for your review - the points you made are helpful for us to improve our manuscript!

---

> > ### Author Response · Authors · 2023-08-30
> > **Response to Reviewer PvGq**
> >
> > Dear reviewer,
> > Thank you again for taking the time to review our work. Let us know if you have any concerns left after our response. We would be happy to discuss any further questions and comments you may have. Please let us know if you have received our responses and if we have successfully addressed your concerns. Thank you once again for your feedback.

---

> > > ### Comment · Reviewer_PvGq · 2023-08-30
> > >
> > > Thanks to the authors for addressing all my concerns

---

### Author Response · Authors · 2023-08-29
**General Response To All Reviewers**

We thank our reviewers for their helpful feedback and their support.

We are glad that all reviewers saw our work as a potentially very impactful contribution to the ML and climate science community (“*This contribution fills a clear need and is obviously relevant to the broad climate change research community.*” [PvGq]; “*...should draw attention from other researchers who are looking to make an impact on this area*” [MSxX]). We appreciate that the reviewers find the dataset and the paper accessible, also for an audience without domain knowledge (“*the paper+appendix quite detailed, which should make it easy for users to understand and interface with*” [MSxX]; “*The ClimateSet data pipeline provided to access the dataset can preprocess the raw data without significant domain knowledge required*” [jcks]) and that they regard the size of the dataset as a new and significant contribution (“*The authors show the benefit of the constructed dataset, e.g. much bigger and practical than previous ones.*” [fdkb]; “*The number (and diversity) of the climate models used could be very important towards building foundation models that can emulate different climate scenarios and is highly relevant to machine learning researchers in the climate science domain.*” [NfhR]).

We found many helpful suggestions and questions in the reviewers' comments and addressed each of them in the individual threads. Here, we summarize the main revisions of the paper conducted in response to the reviews.

We added:
- an overview figure that summarizes the emulation task and we have also expanded the description of the ML task in the main paper;
- a figure showing that different climate models span overall different temperature ranges (considering all SSP scenarios) - the figure shows quantitatively that climate models temperature projections have differences among each other;
- a description of the limitations of the current evaluation metrics;
- clarifications of climate science terms and links to more detailed introductions for such terms.

We adapted:
- the introduction. We rewrote, restructured and condensed the introduction to better introduce terms that may be unfamiliar to the audience,  to avoid repetition and to respond to specific reviewer comments.
- the dataset section slightly to condense the content and avoid repetition;
- the super emulation section to discuss the one-to-many mapping issue raised in the review; we included the updated results of the super emulation experiments and updated the discussion of those results.

We additionally updated our appendix by
- adapting the section on our super emulation setup to include our newest architecture;
- adding GP results and extending their discussion;
- removing the RF description section since we do not report results for RFs;
- restructuring the generalization capabilities sections, including another table, and extending the discussion.

We thank everyone involved in reviewing the paper for their time.

---

### Decision · Program_Chairs · 2023-09-22

**Decision:**

Accept (Poster)

**Comment:**

This paper introduces a unified dataset and benchmark for climate modeling, collating the results from 36 climate models, as well as baseline model results and a pipeline for processing additional climate model data. A couple of reviewers raised good points about multimodality in the output space and the evaluation metrics (e.g. RMSE) falling short in capturing that. This discussion is useful to include in the paper as an avenue for future work in developing the right metrics. Overall, this is a solid contribution towards training large-scale ML models on climate simulations. I recommended acceptance.